# AN EFFECTIVE UNIVERSAL POLYNOMIAL BASIS FOR SPECTRAL GRAPH NEURAL NETWORKS

## ABSTRACT

Spectral Graph Neural Networks (GNNs), also referred to as *graph filters* have gained increasing prevalence for heterophily graphs. Optimal graph filters rely on Laplacian eigendecomposition for Fourier transform. In an attempt to avert the prohibitive computations, numerous polynomial filters by leveraging distinct polynomials have been proposed to approximate the desired graph filters. However, polynomials in the majority of polynomial filters are *predefined* and remain *fixed* across all graphs, failing to accommodate the diverse heterophily degrees across different graphs. To tackle this issue, we first investigate the correlation between polynomial bases of desired graph filters and the degrees of graph heterophily via a thorough theoretical analysis. Afterward, we develop an adaptive heterophily basis by incorporating graph heterophily degrees. Subsequently, we integrate this heterophily basis with the homophily basis, creating a universal polynomial basis *UniBasis*. In consequence, we devise a general polynomial filter *UniFilter*. Comprehensive experiments on both real-world and synthetic datasets with varying heterophily degrees significantly support the superiority of UniFilter, demonstrating the effectiveness and generality of UniBasis, as well as its promising capability as a new method for graph analysis.

## 1 INTRODUCTION

Spectral Graph Neural Networks (GNNs) (Kipf & Welling, 2017), known as *graph filters* have been extensively investigated in recent years due to their superior performance in handling heterophily graphs. Optimal graph filters conduct Laplacian eigendecomposition for Fourier transform. To bypass the computation complexity, existing graph filters leverage various polynomials to approximate the desired filters for graphs with varying heterophily degrees. For example, ChebNet (Defferrard et al., 2016) employs truncated Chebyshev polynomials (Mason & Handscomb, 2002; Hammond et al., 2011) and accomplishes localized spectral filtering. BernNet (He et al., 2021) utilizes Bernstein polynomials (Farouki, 2012) to acquire better controllability and interpretability. Later, Wang & Zhang (2022) propose JacobiConv by exploiting Jacobi polynomial bases (Askey, 1974) with improved generality. Recently, the state-of-the-art (SOTA) graph filter OptBasisGNN (Guo & Wei, 2023) orthogonalizes the polynomial basis to reach the maximum convergence speed.

However, polynomial bases utilized in existing polynomial filters ignore the varying heterophily degrees underlying graphs. Among them, orthogonal bases are proved optimal in terms of convergence speed Wang & Zhang (2022); Guo & Wei (2023). Yet, it demonstrates suboptimal empirical performance on node classification, especially on strong homophily graphs in our experiments (sections 5.1 and 5.3). This scenario arises from the lack of consideration of the homophily property in the construction of the orthonormal basis, rendering it inferior to strong homophily graphs. As we prove in Theorem 1, frequencies of signals filtered by optimal graph filters are proportional to the heterophily degrees. This suggests that ideal polynomial bases are obligated to provide adaptability to the diverse heterophily degrees.

Ergo, a natural question to ask is: **how can we design a universal polynomial basis that encapsulates the graph heterophily degrees?** Inspired, we first establish the relation between the heterophily degree and the frequency of optimal filtered signals (Theorem 1). Subsequently, we explore how the distribution of polynomial bases in Euclidean space affects the basis spectrum (Theorem 3). Based on those insightful findings, we design an adaptive heterophily basis by incorporating het-

erophily degrees of graphs. Eventually, we integrate the heterophily basis and the homophily basis into a universal basis denoted as *UniBasis*. Upon UniBasis, we devise a general polynomial filter called *UniFilter*. For a comprehensive evaluation, we compare UniFilter with 20 baselines on 6 real-world datasets and synthetic datasets with a range of heterophily degrees. The notably superior performance of UniFilter strongly confirms the effectiveness and generality of UniBasis, especially on heterophily graphs. Meanwhile, we demonstrate the spectrum distribution of trained UniBasis on each tested dataset (section 5.2). The experimental results explicitly support the promising capability of UniBasis as a new method for graph analysis with enriched interpretability.

In a nutshell, our contribution can be summarized as: 1) We reveal that underlying polynomials of desired polynomial filters are meant to keep aligned with degrees of graph heterophily; 2) We design a universal polynomial basis UniBasis by incorporating graph heterophily degrees and devise a general graph filter UniFilter; 3) We evaluate UniFilter on both real-world and synthetic datasets against 18 baselines. The remarkable performance of UniFilter strongly confirms the effectiveness and generality of UniBasis, as well as its promising capability as a new method for graph analysis.

## 2 PRELIMINARIES

### 2.1 NOTATIONS AND DEFINITIONS

We represent matrices, vectors, and sets with bold uppercase letters (e.g., $\mathbf{A}$), bold lowercase letters (e.g., $\mathbf{x}$), and calligraphic fonts (e.g., $\mathcal{N}$), respectively. The $i$-th row (resp. column) of matrix $\mathbf{A}$ is represented by $\mathbf{A}[i, \cdot]$ (resp. $\mathbf{A}[\cdot, i]$). We denote $[n] = \{1, 2, \cdots, n\}$.

Let $\mathbf{G} = (\mathcal{V}, \mathcal{E})$ be an undirected and connected graph with node set $|\mathcal{V}| = n$ and edge set $|\mathcal{E}| = m$. Let $\mathbf{X} \in \mathbb{R}^{n \times d}$ be the $d$-dimension feature matrix. For ease of exposition, we employ node notation $u \in \mathcal{V}$ to denote its index, i.e., $\mathbf{X}_u = \mathbf{X}[u, \cdot]$. Let $\mathbf{Y} \in \mathbb{N}^{n \times |\mathcal{C}|}$ be the one-hot label matrix, i.e., $\mathbf{Y}[u, i] = 1$ if node $u$ belongs to class $\mathcal{C}_i$ for $i \in \{1, 2, \cdots, |\mathcal{C}|\}$, where $\mathcal{C}$ is the set of node labels. The set of direct (one-hop) neighbors of node $u \in \mathcal{V}$ is denoted as $\mathcal{N}_u$ with degree $d_u = |\mathcal{N}_u|$. The adjacency matrix of $\mathbf{G}$ is denoted as $\mathbf{A} \in \mathbb{R}^{n \times n}$ that $\mathbf{A}[u, v] = 1$ if edge $\langle u, v \rangle \in \mathcal{E}$; otherwise $\mathbf{A}[u, v] = 0$. $\mathbf{D} \in \mathbb{R}^{n \times n}$ is the diagonal degree matrix of $\mathbf{G}$ with $\mathbf{D}[u, u] = d_u$. Let $\mathbf{L}$ be the normalized Laplacian matrix of graph $\mathbf{G}$ defined as $\mathbf{L} = \mathbf{I} - \mathbf{D}^{-\frac{1}{2}} \mathbf{A} \mathbf{D}^{-\frac{1}{2}}$ where $\mathbf{I}$ is the identity matrix and $\hat{\mathbf{L}}$ be the normalized Laplacian matrix of $\mathbf{G}$ with self-loops as $\hat{\mathbf{L}} = \mathbf{I} - \tilde{\mathbf{D}}^{-\frac{1}{2}} \tilde{\mathbf{A}} \tilde{\mathbf{D}}^{-\frac{1}{2}}$ where $\tilde{\mathbf{D}} = \mathbf{D} + \mathbf{I}$ and $\tilde{\mathbf{A}} = \mathbf{A} + \mathbf{I}$.

### 2.2 SPECTRAL GRAPH FILTERS

In general, the eigendecomposition of the Laplacian matrix is denoted as $\mathbf{L} = \mathbf{U} \mathbf{\Lambda} \mathbf{U}^{\top}$, where $\mathbf{U}$ is the matrix of eigenvectors and $\mathbf{\Lambda} = \text{diag}[\lambda_1, \cdots, \lambda_n]$ is the diagonal matrix of eigenvalues. Eigenvalues $\lambda_i$ for $i \in [n]$ mark the *frequency* and the eigenvalue set $\{\lambda_1, \cdots, \lambda_n\}$ is the *graph spectrum*. Without loss of generality, we assume $0 = \lambda_1 \leq \lambda_2 \leq \cdots \leq \lambda_n \leq 2$. When applying a spectral graph filter on graph signal $\mathbf{x} \in \mathbb{R}^n$, the process involves the following steps. First, the graph Fourier operator $\mathcal{F}(\mathbf{x}) = \mathbf{U}^{\top} \mathbf{x}$ projects the graph signal $\mathbf{x}$ into the spectral domain. Subsequently, a spectral filtering function $g_{\mathbf{w}}(\cdot)$ parameterized by $\mathbf{w} \in \mathbb{R}^n$ is employed on the derived spectrum. Eventually, the filtered signal is transformed back via the inverse graph Fourier transform operator $\mathcal{F}^{-1}(\mathbf{x}) = \mathbf{U} \mathbf{x}$. The process is formally expressed as

$$\mathcal{F}^{-1}(\mathcal{F}(g_{\mathbf{w}}) \odot \mathcal{F}(\mathbf{x})) = \mathbf{U} g_{\mathbf{w}}(\mathbf{\Lambda}) \mathbf{U}^{\top} \mathbf{x} = \mathbf{U} \, \text{diag}(g_{\mathbf{w}}(\lambda_1), \cdots, g_{\mathbf{w}}(\lambda_n)) \mathbf{U}^{\top} \mathbf{x}, \quad (1)$$

where $\odot$ is the Hadamard product.

In particular, spectral graph filters enhance signals in specific frequency ranges and suppress the signals in the rest parts according to objective functions. For node classification, homophily graphs are prone to contain low-frequency signals whilst heterophily graphs likely own high-frequency signals. In order to quantify the heterophily degrees of graphs, numerous homophily metrics have been introduced, e.g., *edge homophily* (Zhu et al., 2020), *node homophily* (Pei et al., 2020), *class homophily* (Lim et al., 2021; Luan et al., 2021), and a recent *adjusted homophily* (Platonov et al., 2022). By following the literature of spectral graph filters (Zhu et al., 2020; Lei et al., 2022), we adopt edge homophily in this work, explained as follows.

**Table 1: Polynomial Graph Filters**

| | Poly. Basis | Graph Filter $g_{\mathbf{w}}(\lambda)$ | Prop. Matrix $\mathbf{P}$ |
|---|---|---|---|
| ChebNet (Defferrard et al., 2016) | Chebyshev | $\sum_{k=0}^{K} \mathbf{w}_k T_k(\hat{\lambda})$ | $2\mathbf{L}/\lambda_{max} - \mathbf{I}$ |
| GPR-GNN (Chien et al., 2021) | Monomial | $\sum_{k=0}^{K} \mathbf{w}_k (1 - \tilde{\lambda})^k$ | $\mathbf{I} - \hat{\mathbf{L}}$ |
| BernNet (He et al., 2021) | Bernstein | $\sum_{k=0}^{K} \frac{\mathbf{w}_k}{2^K} \binom{K}{k}(2 - \lambda)^{K-k}\lambda^k$ | $\mathbf{I} - \frac{\mathbf{L}}{2}$ |
| JacobiConv (Wang & Zhang, 2022) | Jacobi | $\sum_{k=0}^{K} \mathbf{w}_k \mathbf{P}_k^{a,b}(1 - \lambda)$ | $\mathbf{I} - \mathbf{L}$ |
| OptBasisGNN (Guo & Wei, 2023) | Orthonormal | — | $\mathbf{I} - \mathbf{L}$ |

**Definition 1 (Homophily Ratio $h$)** *Given a graph $\mathbf{G} = (\mathcal{V}, \mathcal{E})$ and its label matrix $\mathbf{Y}$, the homophily ration $h$ of $\mathbf{G}$ is the fraction of edges with two end nodes from the same class, i.e., $h = \frac{|\{\langle u,v \rangle \in \mathcal{E}:\ \mathbf{y}_u = \mathbf{y}_v\}|}{|\mathcal{E}|}$.*

Besides the homophily metrics for *categorical* node labels, the similarity of *numerical* node signals can also be measured via *Dirichlet Enenrgy* (Zhou et al., 2021; Karhadkar et al., 2023). Specifically, we customize the metric to node signals $\mathbf{x} \in \mathbb{R}^n$ and propose *spectral signal frequency* as follows.

**Definition 2 (Spectral Signal Frequency $f$)** *Consider a graph $\mathbf{G} = (\mathcal{V}, \mathcal{E})$ with $n$ nodes and Laplacian matrix $\mathbf{L}$. Given a normalized feature signal $\mathbf{x} \in \mathbb{R}^n$, the spectral signal frequency $f(\mathbf{x})$ on $\mathbf{G}$ is defined as $f(\mathbf{x}) = \frac{\mathbf{x}^\top \mathbf{L} \mathbf{x}}{2}$.*

By nature of Dirichlet energy, spectral signal frequency $f(x)$ quantifies the discrepancy of signal $\mathbf{x}$ on graph $\mathbf{G}$. For $f(x)$, it holds that

**Lemma 2.1** *For any normalized feature signal $\mathbf{x} \in \mathbb{R}^n$ on graph $\mathbf{G}$, the spectral signal frequency $f(\mathbf{x}) \in [0,1]$ holds.*

## 3 REVISITING POLYNOMIAL GRAPH FILTERS

Optimal graph filters require eigendecomposition on the Laplacian matrix at the cost of $O(n^3)$. To bypass the high computation overhead, a plethora of polynomial graph filters (Defferrard et al., 2016; Chien et al., 2021; He et al., 2021; Wang & Zhang, 2022; He et al., 2022; Guo & Wei, 2023) have been proposed to approximate optimal graph filters by leveraging distinct polynomials. Table 1 summarizes several such polynomial graph filters, including adopted polynomials, graph filter functions, and propagation matrices if applicable.

By identifying the appropriate matrix $\mathbf{P}$, those polynomial filters applied on graph signal $\mathbf{x} \in \mathbb{R}^n$ can be equally expressed as

$$\mathbf{z} = \sum_{k=0}^{K} \mathbf{w}_k \mathbf{P}^k \cdot \mathbf{x}, \tag{2}$$

where $K$ is the length of polynomial basis, $\mathbf{w} \in \mathbb{R}^{K+1}$ is the learnable weight vector, and $\mathbf{z} \in \mathbb{R}^n$ is the final representation. For example, He et al. (2021) utilize Bernstein polynomial and propose polynomial filter BernNet as $\mathbf{z} = \sum_{k=0}^{K} \frac{\mathbf{w}_k'}{2^K} \binom{K}{k}(2\mathbf{I} - \mathbf{L})^{K-k}\mathbf{L}^k\mathbf{x}$. By setting $\mathbf{P} = \mathbf{I} - \frac{\mathbf{L}}{2}$ as the propagation matrix and rearranging the expression, an equivalent formulation is expressed as $\mathbf{z} = \sum_{k=0}^{K} \mathbf{w}_k \left(\mathbf{I} - \frac{\mathbf{L}}{2}\right)^k \mathbf{x}$ where $\mathbf{w}_k = \sum_{i=0}^{k} \mathbf{w}_{k-i}'\binom{K}{K-i}\binom{K-i}{k-i}(-1)^{k-i}$ is the learnable parameter.

In particular, vectors $\mathbf{P}^k\mathbf{x}$ in Equation (2) for $k \in \{0, 1, \cdots, K\}$ collectively constitute a signal basis $\{\mathbf{P}^0\mathbf{x}, \mathbf{P}^1\mathbf{x}, \cdots, \mathbf{P}^K\mathbf{x}\}$. Spectral graph filters attempt to learn a weighted combination of signal bases, aiming to systematically produce node representations for nodes from graphs with varying heterophily degrees for label prediction. From the spectral perspective, spectral filters essentially execute filtering operations on the spectrum $\{f(\mathbf{P}^0\mathbf{x}), f(\mathbf{P}^1\mathbf{x}), \cdots, f(\mathbf{P}^K\mathbf{x})\}$ in order to approximate the frequencies of label signals $\mathbf{Y}$. Meanwhile, label signal frequencies are closely correlated with the homophily ratio $h$. To formally depict the correlation between the filtered signal $\sum_{k=0}^{K} \mathbf{w}_k \mathbf{P}^k\mathbf{x}$ and homophily ratio $h$, we establish a theorem as follows.

**Theorem 1** *Given a connected graph $\mathbf{G} = (\mathcal{V}, \mathcal{E})$ with homophily ratio $h$, consider an optimal polynomial filter $\mathrm{F}(\mathbf{w}) = \sum_{k=0}^{K} \mathbf{w}_k \mathbf{P}^k$ with propagation matrix $\mathbf{P}$ and weights $\mathbf{w} \in \mathbb{R}^{K+1}$ toward*

$\mathbf{G}$ *for node classification. Given a feature signal* $\mathbf{x} \in \mathbb{R}^n$, *the spectral frequency* $f(\sum_{k=0}^{K} \mathbf{w}_k \mathbf{P}^k \mathbf{x})$ *is proportional to* $1-h$.

Theorem 1 uncovers the critical role of graph homophily ratios when generating desired node representations. Intuitively, ideal signal bases are obligated to consider different heterophily degrees for various graphs. However, the majority of existing polynomial filters exploit predefined polynomials, ignoring the corresponding homophily ratios.

## 4 UNIVERSAL POLYNOMIAL BASIS FOR GRAPH FILTERS

### 4.1 THEORETICAL ANALYSIS OF HOMOPHILY BASIS

Conventional GNN models (Kipf & Welling, 2017; Hamilton et al., 2017; Klicpera et al., 2019a) employ homophily as a strong inductive bias (Lim et al., 2021). To aggregate information within $K$ hops, graph signal $\mathbf{x}$ are propagated to $K$-hop neighbors via propagation matrix $\mathbf{P} = \mathbf{I} - \mathbf{L}$, yielding *homophily basis* $\{\mathbf{x}, \mathbf{Px}, \cdots, \mathbf{P}^K\mathbf{x}\}$. To elucidate how the homophily basis accommodates homophily graphs, we establish the following Theorem.

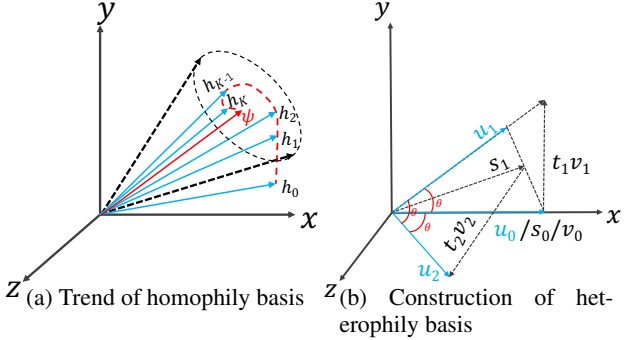

(a) Trend of homophily basis  (b) Construction of heterophily basis

**Figure 1: Illustration of homophily and heterophily bases**

**Theorem 2** *Given a propagation matrix* $\mathbf{P}$ *and graph signal* $\mathbf{x}$, *consider an infinite homophily basis* $\{\mathbf{x}, \mathbf{Px}, \cdots, \mathbf{P}^k\mathbf{x}, \mathbf{P}^{k+1}\mathbf{x}, \cdots\}$. *It holds that (i) as the exponent* $k$ *increases, the angle* $\arccos\left(\frac{\mathbf{P}^k\mathbf{x} \cdot \mathbf{P}^{k+1}\mathbf{x}}{\|\mathbf{P}^k\mathbf{x}\|\|\mathbf{P}^{k+1}\mathbf{x}\|}\right)$ *is progressively smaller, and (ii)* $\lim_{K \to \infty} \arccos\left(\frac{\mathbf{P}^K\mathbf{x} \cdot \psi}{\|\mathbf{P}^K\mathbf{x}\|\|\psi\|}\right) = 0$ *where* $\psi = \mathbf{D}^{\frac{1}{2}}\mathbf{1}$.

The homophily basis exhibits *growing similarity* and *asymptotic convergence* for the purpose of capturing homophily signals, thus resulting in the *over-smoothing issue*. For better visualization, Figure 1a simply illustrates the homophily basis $\{\mathbf{h}_0, \mathbf{h}_1, \mathbf{h}_2, \cdots, \mathbf{h}_{K-1}, \mathbf{h}_K, \cdots\}$ gradually converges to $\psi$ in 3-dimension Euclidean space.

### 4.2 ADAPTIVE HETEROPHILY BASIS

As discussed aforementioned, desired signal bases are expected to conform to homophily ratios. A natural question is: *how can we apply homophily ratios in a sensible manner when designing signal bases without involving graph signals or structures?* To answer this question, we initially explore the correlation between the basis distribution in Euclidean space and the basis frequency on regular graphs.

**Theorem 3** *Consider a regular graph* $\mathbf{G}$, *a random basis signal* $\mathbf{x} \in \mathbb{R}^n$, *and a normalized all-ones vector* $\phi \in \mathbb{R}^n$ *with frequency* $f(\phi) = 0$. *Suppose* $\theta := \arccos(\phi \cdot \mathbf{x})$ *denotes the angle formed by* $\mathbf{x}$ *and* $\phi$. *It holds that the expectation of spectral signal frequency* $\mathbb{E}_{\mathbf{G} \sim \mathcal{G}}[f(\mathbf{x})]$ *over the randomness of* $\mathbf{G}$ *is monotonically increasing with* $\theta$ *for* $\theta \in [0, \frac{\pi}{2})$.

Theorem 3 reveals the correlation between the expected frequency of the signal basis and its relative position to the 0-frequency vector $\phi$ on regular graphs. This fact implicitly suggests that we may take the angles (relative position) between two basis vectors into consideration when aiming to achieve the desired basis spectrum on general graphs. Meanwhile, Theorem 2 discloses the growing similarity and asymptotic convergence phenomenon within the homophily basis. To mitigate this over-smoothing issue, we can intuitively enforce all pairs of basis vectors to form an appropriate angle of $\theta \in [0, \frac{\pi}{2}]$. Pertaining to this, Theorem 1 proves the spectral frequency of ideal signals proportional to $1 - h$, aligning with the homophily ratios of the underlying graphs. By leveraging

the monotonicity property proved in Theorem 3, we empirically set the $\theta := \frac{\pi}{2}(1-h)$. Consequently, a signal basis capable of capturing the heterophily degrees of graphs is derived, formally denoted as *heterophily basis*.

Consider to construct a heterophily basis with a length of $K+1$. The procedure of computing heterophily basis is outlined in Algorithm 1 and illustrated in Figure 1b. To start with, we normalize the input signal $\mathbf{x}$ as the initial signal $\mathbf{u}_0$ and set $\theta := \frac{(1-h)\pi}{2}$. In order to manipulate the formed angles between signal vectors, we forge an orthonormal basis, denoted as $\{\mathbf{v}_0, \mathbf{v}_1, \cdots, \mathbf{v}_K\}$ where $\mathbf{v}_0$ is initialized as $\mathbf{u}_0$. In particular, at the $k$-th iteration for $k \in [1, K]$, we set $\mathbf{v}_k = \mathbf{P}\mathbf{v}_{k-1}$ where $\mathbf{P} = \mathbf{I} - \mathbf{L}$ is the propagation matrix. Subsequently, $\mathbf{v}_k$ is calculated as $\mathbf{v}_k := \mathbf{v}_k - (\mathbf{v}_k^\top \mathbf{v}_{k-1})\mathbf{v}_{k-1} - (\mathbf{v}_k^\top \mathbf{v}_{k-2})\mathbf{v}_{k-2}$ as per the *three-term recurrence* Theorem (Gautschi, 2004; Liesen & Strakos, 2013; Guo & Wei, 2023). Meanwhile, signal vector $\mathbf{u}_k$ is set as $\mathbf{u}_k := \frac{\mathbf{s}_{k-1}}{k}$ where $\mathbf{s}_{k-1} := \sum_{i=0}^{k-1} \mathbf{u}_i$. Subsequently, $\mathbf{u}_k$ is updated as $\mathbf{u}_k := \frac{\mathbf{u}_k + t_k \mathbf{v}_k}{\|\mathbf{u}_k + t_k \mathbf{v}_k\|}$ where $t_k$ is

$$t_k = \sqrt{\left(\frac{\mathbf{s}_{k-1}^\top \mathbf{u}_{k-1}}{k\cos(\theta)}\right)^2 - \frac{(k-1)\cos(\theta)+1}{k}}. \tag{3}$$

As a result, the final vector set $\{\mathbf{u}_0, \mathbf{u}_1, \cdots, \mathbf{u}_K\}$ is returned as the heterophily basis. The desired property of the heterophily basis is proved in the following Theorem. Detailed proofs are presented in Appendix A.1.

---

**Algorithm 1:** Heterophily Basis

**Input:** Graph $\mathbf{G}$, propagation matrix $\mathbf{P}$, input feature signal $\mathbf{x}$, hop $K$, estimated homophily ratio $\hat{h}$

**Output:** Heterophily basis $\{\mathbf{u}_0, \mathbf{u}_1, \cdots, \mathbf{u}_K\}$

1   $\mathbf{u}_0 \leftarrow \frac{\mathbf{x}}{\|\mathbf{x}\|}, \mathbf{v}_0 \leftarrow \mathbf{u}_0, \mathbf{v}_{-1} \leftarrow \mathbf{0}, \mathbf{s}_0 \leftarrow \mathbf{u}_0, \theta \leftarrow \frac{(1-\hat{h})\pi}{2}$;

2   **for** $k \leftarrow 1$ **to** $K$ **do**

3      $\mathbf{v}_k \leftarrow \mathbf{P}\mathbf{v}_{k-1}$;

4      $\mathbf{v}_k \leftarrow \mathbf{v}_k - (\mathbf{v}_k^\top \mathbf{v}_{k-1})\mathbf{v}_{k-1} - (\mathbf{v}_k^\top \mathbf{v}_{k-2})\mathbf{v}_{k-2}$;

5      $\mathbf{v}_k \leftarrow \frac{\mathbf{v}_k}{\|\mathbf{v}_k\|}, \mathbf{u}_k \leftarrow \frac{\mathbf{s}_{k-1}}{k}$;

6      $t_k$ is calculated as in Equation (3);

7      $\mathbf{u}_k \leftarrow \frac{\mathbf{u}_k + t_k \mathbf{v}_k}{\|\mathbf{u}_k + t_k \mathbf{v}_k\|}, \mathbf{s}_k \leftarrow \mathbf{s}_{k-1} + \mathbf{u}_k$;

8   **return** $\{\mathbf{u}_0, \mathbf{u}_1, \cdots, \mathbf{u}_K\}$;

---

**Theorem 4** *Consider a heterophily basis $\{\mathbf{u}_0, \mathbf{u}_1, \cdots \mathbf{u}_K\}$ constructed from Algorithm 1 for graphs with homophily ratio $h$. It holds that* $\mathbf{u}_i \cdot \mathbf{u}_j = \begin{cases} \cos(\frac{(1-h)\pi}{2}) & \text{if } i \neq j \\ 1 & \text{if } i = j \end{cases}$ *for* $\forall i, j \in \{0, 1, \cdots, K\}$.

**Homophily ratio estimation.** The exact homophily ratio $h$ relies on the label set of the entire graph and thus is normally unavailable. To address this issue, we estimate $h$ through labels of training data, denoted as $\hat{h}$. Appendix A.3 presents the experimental results of the homophily ratio estimation, which signifies that a qualified homophily ratio can be effectively estimated via training data.

**Time complexity.** In the $k$-th iteration, it takes $O(m + n)$ to calculate the orthonormal basis and $O(n)$ to update $\mathbf{u}_k$. Therefore, the total time complexity of Algorithm 1 is $O(K(m+n))$, i.e., linear to propagation hops and input graph sizes.

### 4.3 UNIVERSAL POLYNOMIAL BASIS AND GRAPH FILTER

The heterophily basis employs fixed angle $\theta := \frac{(1-\hat{h})\pi}{2}$ associated with heterophily degrees, effectively encapsulating the heterophily of graphs. However, it can be restrictive by nature when handling strong homophily graphs with homophily ratios $h$ close to 1. To tackle graphs ranging from strong homophily to strong heterophily, we intuitively introduce a hyperparameter $\tau \in [0, 1]$

and merge the homophily basis and heterophily basis into a universal polynomial $\tau\mathbf{P}^k\mathbf{x}+(1-\tau)\mathbf{u}_k$, referred to as *UniBasis*. As a consequence, a general polynomial filter *UniFilter* is proposed as

$$\mathbf{z} = \sum_{k=0}^{K} \mathbf{w}_k(\tau\mathbf{P}^k\mathbf{x} + (1 - \tau)\mathbf{u}_k) \tag{4}$$

with learnable weight vector $\mathbf{w} \in \mathbb{R}^{K+1}$.

**Convergence Discussion.** The convergence speed of $\mathbf{P}^K\mathbf{x}$ to $\psi$ in Theorem 2 is affected by the *Cheerger constant* (Chung & Graham, 1997) of the underlying graphs. In general, dense graphs with larger Cheeger constant exhibit more rapid convergence while it is contrary for sparse graphs. Meanwhile, the rate of basis approximation convergence is determined by the *condition number* of the Hessian matrix (Wright et al., 1999; Boyd et al., 2004). It is known that orthogonal polynomial bases achieve the maximum convergence rate (Wang & Zhang, 2022; Guo & Wei, 2023). Yet it is essential to emphasize that orthonormal bases do not consistently yield empirically superior node representations, as verified in Sections 5.1 and 5.3.

## 5 EXPERIMENTS

**Datasets.** We evaluate the performance of UniFilter on 6 real-world datasets with varied homophily ratios. Specifically, the three citation networks (Sen et al., 2008), i.e., Cora, Citeseer, and Pubmed, are homophily graphs with homophily ratios 0.81, 0.73, and 0.80 respectively; the two Wikipedia graphs, i.e., Chameleon and Squirrel and the Actor co-occurrence graph from WebKB3 (Pei et al., 2020) are heterophily graphs with homophily ratios 0.22, 0.23, and 0.22 respectively. Dataset details are presented in Table 4 in Appendix A.2.

**Baselines.** We compare UniFilter with 20 baselines in two categories, i.e., *polynomial filters* and *model-optimized methods*. Specifically, polynomial filters employ various polynomials to approximate the optimal graph filters, including monomial SGC (Wu et al., 2019), SIGN (Frasca et al., 2020), ASGC Chanpuriya & Musco (2022), GPR-GNN (Chien et al., 2021), and EvenNet (Lei et al., 2022), Chebyshev polynomial ChebNet (Defferrard et al., 2016) and its improved version ChebNetII (He et al., 2022), Bernstein polynomial BernNet (He et al., 2021), Jacobi polynomial JacobiConv (Wang & Zhang, 2022), the orthogonal polynomial OptBasisGNN (Guo & Wei, 2023) and learnable basis Specformer (Bo et al., 2023). In contrast, model-optimized methods optimize the architecture for improved node representations, including GCN (Kipf & Welling, 2017), GCNII (Chen et al., 2020), GAT (Velickovic et al., 2018), MixHop (Abu-El-Haija et al., 2019), H$_2$GCN (Zhu et al., 2020), LINKX (Lim et al., 2021), WRGAT (Suresh et al., 2021), ACM-GCN (Luan et al., 2022), and GloGNN++ (Li et al., 2022).

**Experiment Settings.** There are two common data split settings 60%/20%/20% and 48%/32%/20% for train/validation/test in the literature. Specifically, the polynomial filters are mostly tested in the previous setting (He et al., 2021; Wang & Zhang, 2022; Guo & Wei, 2023; Bo et al., 2023) while the model-optimized methods are normally evaluated in the latter[1] (Zhu et al., 2020; Li et al., 2022; Song et al., 2023).

### 5.1 NODE CLASSIFICATION PERFORMANCE

Table 2 and Table 3 present the results of UniFilter compared with existing polynomial filters and model-optimized methods for node classification respectively. For ease of exposition, we highlight the *highest* accuracy score in bold and underline the *second highest* score for each dataset.

As shown, our method UniFilter consistently achieves the highest accuracy scores on both the homophily datasets and heterophily datasets, except in one case on Actor in Table 2. UniFilter exhibits explicit performance advantages over both SOTA polynomial filter Specformer and SOTA model-optimized method GloGNN++ for the majority of cases. In particular, the performance improvements are remarkably significant on the two heterophily datasets Chameleon and Squirrel. Specifically, the corresponding performance gains reach up to 1.03% and 2.76% in Table 2 and 2.45% and 6.38% in Table 3 respectively. It is worth mentioning that the computation time of UniBasis is linear

---

[1]Please note that those model-optimized methods reuse the public data splits from Pei et al. (2020) which are actually in the splits of 48%/32%/20% in the implementation.

**Table 2: Accuracy (%) compared with polynomial filters**

| Methods | Cora | Citeseer | Pubmed | Actor | Chameleon | Squirrel |
|---|---|---|---|---|---|---|
| SGC | $86.83 \pm 1.28$ | $79.65 \pm 1.02$ | $87.14 \pm 0.90$ | $34.46 \pm 0.67$ | $44.81 \pm 1.20$ | $25.75 \pm 1.07$ |
| SIGN | $87.70 \pm 0.69$ | $80.14 \pm 0.87$ | $89.09 \pm 0.43$ | $41.22 \pm 0.96$ | $60.92 \pm 1.45$ | $45.59 \pm 1.40$ |
| ASGC | $85.35 \pm 0.98$ | $76.52 \pm 0.36$ | $84.17 \pm 0.24$ | $33.41 \pm 0.80$ | $71.38 \pm 1.06$ | $57.91 \pm 0.89$ |
| GPR-GNN | $88.54 \pm 0.67$ | $80.13 \pm 0.84$ | $88.46 \pm 0.31$ | $39.91 \pm 0.62$ | $67.49 \pm 1.38$ | $50.43 \pm 1.89$ |
| EvenNet | $87.77 \pm 0.67$ | $78.51 \pm 0.63$ | $90.87 \pm 0.34$ | $40.36 \pm 0.65$ | $67.02 \pm 1.77$ | $52.71 \pm 0.85$ |
| ChebNet | $87.32 \pm 0.92$ | $79.33 \pm 0.57$ | $87.82 \pm 0.24$ | $37.42 \pm 0.58$ | $59.51 \pm 1.25$ | $40.81 \pm 0.42$ |
| ChebNetII | $88.71 \pm 0.93$ | $80.53 \pm 0.79$ | $88.93 \pm 0.29$ | $41.75 \pm 1.07$ | $71.37 \pm 1.01$ | $57.72 \pm 0.59$ |
| BernNet | $88.51 \pm 0.92$ | $80.08 \pm 0.75$ | $88.51 \pm 0.39$ | $41.71 \pm 1.12$ | $68.53 \pm 1.68$ | $51.39 \pm 0.92$ |
| JacobiConv | $88.98 \pm 0.72$ | $80.78 \pm 0.79$ | $89.62 \pm 0.41$ | $41.17 \pm 0.64$ | $74.20 \pm 1.03$ | $57.38 \pm 1.25$ |
| OptBasisGNN | $87.00 \pm 1.55$ | $80.58 \pm 0.82$ | $90.30 \pm 0.19$ | $\mathbf{42.39 \pm 0.52}$ | $74.26 \pm 0.74$ | $63.62 \pm 0.76$ |
| Specformer | $88.57 \pm 1.01$ | $81.49 \pm 0.94$ | $87.73 \pm 0.58$ | $41.93 \pm 1.04$ | $74.72 \pm 1.29$ | $64.64 \pm 0.81$ |
| UniFilter | $\mathbf{89.49 \pm 1.35}$ | $\mathbf{81.39 \pm 1.32}$ | $\mathbf{91.44 \pm 0.50}$ | $40.84 \pm 1.21$ | $\mathbf{75.75 \pm 1.65}$ | $\mathbf{67.40 \pm 1.25}$ |

**Table 3: Accuracy (%) compared with model-optimized methods**

| Methods | Cora | Citeseer | Pubmed | Actor | Chameleon | Squirrel |
|---|---|---|---|---|---|---|
| GCN | $86.98 \pm 1.27$ | $76.50 \pm 1.36$ | $88.42 \pm 0.50$ | $27.32 \pm 1.10$ | $64.82 \pm 2.24$ | $53.43 \pm 2.01$ |
| GCNII | $88.37 \pm 1.25$ | $77.33 \pm 1.48$ | $90.15 \pm 0.43$ | $37.44 \pm 1.30$ | $63.86 \pm 3.04$ | $38.47 \pm 1.58$ |
| GAT | $87.30 \pm 1.10$ | $76.55 \pm 1.23$ | $86.33 \pm 0.48$ | $27.44 \pm 0.89$ | $60.26 \pm 2.50$ | $40.72 \pm 1.55$ |
| MixHop | $87.61 \pm 0.85$ | $76.26 \pm 1.33$ | $85.31 \pm 0.61$ | $32.22 \pm 2.34$ | $60.50 \pm 2.53$ | $43.80 \pm 1.48$ |
| $H_2$GCN | $87.87 \pm 1.20$ | $77.11 \pm 1.57$ | $89.49 \pm 0.38$ | $35.70 \pm 1.00$ | $60.11 \pm 2.15$ | $36.48 \pm 1.86$ |
| LINKX | $84.64 \pm 1.13$ | $73.19 \pm 0.99$ | $87.86 \pm 0.77$ | $36.10 \pm 1.55$ | $68.42 \pm 1.38$ | $61.81 \pm 1.80$ |
| WRGAT | $88.20 \pm 2.26$ | $76.81 \pm 1.89$ | $88.52 \pm 0.92$ | $36.53 \pm 0.77$ | $65.24 \pm 0.87$ | $48.85 \pm 0.78$ |
| ACM-GCN | $87.91 \pm 0.95$ | $77.32 \pm 1.70$ | $90.00 \pm 0.52$ | $36.28 \pm 1.09$ | $66.93 \pm 1.85$ | $54.40 \pm 1.88$ |
| GloGNN++ | $88.33 \pm 1.09$ | $77.22 \pm 1.78$ | $89.24 \pm 0.39$ | $37.70 \pm 1.40$ | $71.21 \pm 1.84$ | $57.88 \pm 1.76$ |
| UniFilter | $\mathbf{89.12 \pm 0.87}$ | $\mathbf{80.28 \pm 1.31}$ | $\mathbf{90.19 \pm 0.41}$ | $\mathbf{37.79 \pm 1.11}$ | $\mathbf{73.66 \pm 2.44}$ | $\mathbf{64.26 \pm 1.46}$ |

to graph sizes and propagation hops. The superior performance of UniFilter strongly confirms the superb effectiveness and generality of UniBasis.

## 5.2 SPECTRUM DISTRIBUTION OF DATASETS

The superior performance of UniFilter explicitly implies the superb capability of UniBasis to capture the spectral characteristics of graphs. For better demonstration, we first calculate the *spectral signal frequency* of each basis vector for all $d$-dimensions, resulting in $d$ spectrum of length $K + 1$. We then average the spectrum and associate it with the learned weights $\mathbf{w}$ accordingly, where weights $\mathbf{w} \in \mathbb{R}^{K+1}$ of UniBasis are trained for each dataset. The spectrum distributions of trained UniBasis for the 6 datasets are plotted in Figure 2.

Recall that signals in specific frequencies with weights in large absolute are enhanced while signals with small weights are suppressed. As displayed, the majority of signals of the three homophily datasets lie within the relatively low-frequency intervals as expected, e.g., $[0.3, 0.5]$. We also observe some minor high-frequency information which also provides insightful information for node classification (Klicpera et al., 2019b; Chen et al., 2019; Balcilar et al., 2020). On the contrary, UniBasis of the three heterophily datasets intends to remove low-frequency signals with negative weights and preserve high-frequency information. The distinct spectrum distributions of UniBasis disclose the unique spectral characteristics of each dataset. Those results manifest the capability of UniBasis as a new method to analyze graphs with varying heterophily degrees in the spectral domain with enriched interpretability.

## 5.3 ABLATION STUDY

**Universality of UniBasis.** We compare UniFilter with three of its variants using distinct polynomial bases in order to verify the effectiveness of UniBasis. To this end, we alter UniFilter by changing UniBasis into 1) a filter simply using the heterophily basis (setting $\tau = 0$) denoted as HetFilter, 2) a filter simply using the homophily basis (setting $\tau = 1$) denoted as HomFilter, and 3) a filter using the orthonormal basis (adopting $\{\mathbf{v}_0, \mathbf{v}_1, \cdots, \mathbf{v}_k\}$) denoted as OrtFilter. For easy control, we generate

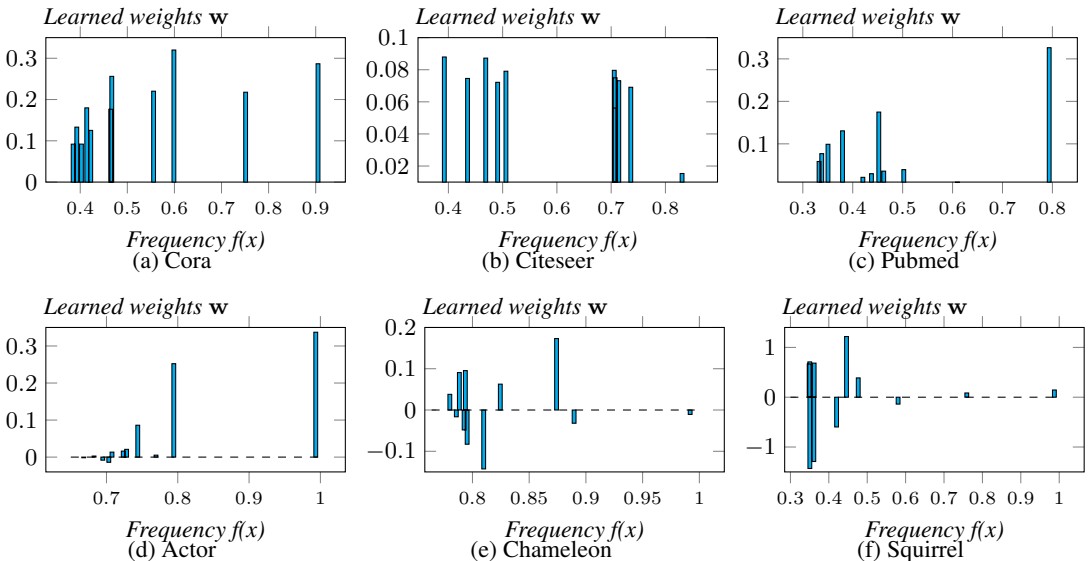

Figure 2: Spectrum distribution of trained UniBasis.

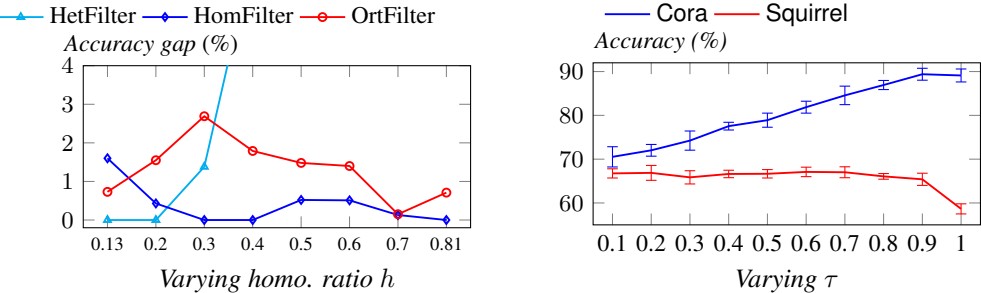

Figure 3: Accuracy gaps of the three variants from UniFilter on $\mathbf{G}_s$ across varying $h$.

Figure 4: Accuracy (%) with varying $\tau$.

a synthetic dataset $\mathbf{G}_s$ by adopting the graph structure and label set of Cora. W.l.o.g., we generate a random one-hot feature vector in $100$ dimensions for each node in $\mathbf{G}_s$. To vary homophily ratios of $\mathbf{G}_s$, we permute nodes in a random sequence and randomly reassign node labels progressively, resulting in homophily ratios in $\{0.13^2, 0.20, 0.30, 0.40, 0.50, 0.60, 0.70, 0.81\}$ accordingly.

The performance advantage gaps of UniFilter over the three variants are presented in Figure 3. We omit the results of HetFilter beyond $h \geq 0.3$ since the corresponding performance gaps become significantly larger, which is as expected since the heterophily basis is incapable of tackling homophily graphs. In particular, the performance advantage of UniFilter over HomFilter gradually decreases as $h$ grows larger. Contrarily, performance gaps of OrtFilter from UniFilter peak at $h = 0.3$ with a notable shortfall and then erratically decrease, ending with an accuracy gap of $0.71\%$ at $h = 0.81$. The fluctuation of OrtFilter states the inferiority of orthonormal basis over UniBasis.

**Sensitivity of $\tau$.** To explore the sensitivity of UniFilter towards the hyperparameter $\tau$, we vary $\tau$ in $\{0, 0.1, \cdots, 0.9, 1\}$ and test UniFilter on the strong homophily dataset Cora and the strong heterophily dataset Squirrel. Figure 4 plots the performance development along varying $\tau$. As displayed, UniFilter prefers the homophily basis on Cora, and the performance peaks at $\tau = 0.9$. On the contrary, the performance of UniFilter slightly fluctuates when $\tau \leq 0.7$ and then moderately

---

2Note that this is the smallest homophily ratio we can possibly acquire by random reassignments.

decreases along the increase of $\tau$ on Squirrel. When $\tau = 1.0$, the accuracy score drops sharply since only the homophily basis is utilized in this scenario.

## 6 RELATED WORK

**Polynomial filters.** As the seminal work, ChebNet (Defferrard et al., 2016) utilizes a $K$-order truncated Chebyshev polynomials (Mason & Handscomb, 2002; Hammond et al., 2011) and provides $K$-hop localized filtering capability. GPR-GNN (Chien et al., 2021) simply adopts monomials instead and applies the generalized PageRank (Li et al., 2019) scores as the coefficients to measure node proximity. In comparison, SGC (Wu et al., 2019) simplifies the propagation by keeping only the $K$-th order polynomial and removing nonlinearity. ASGC Chanpuriya & Musco (2022) simplifies the graph convolution operation by calculating a trainable Krylov matrix so as to adapt various heterophily graphs, which, however, is suboptimal as demonstrated in our experiments. To enhance controllability and interpretability, BernNet (He et al., 2021) employs nonnegative Bernstein polynomials as the basis. Later, Wang & Zhang (2022) examine the expressive power of existing polynomials and propose JacobiConv by leveraging Jacobi polynomial (Askey, 1974), achieving better adaptability to underlying graphs. Subsequently, He et al. (2022) revisit ChebNet and pinpoint the over-fitting issue in Chebyshev approximation. To address the issue, they turn to Chebyshev interpolation and propose ChebNetII. Recently, polynomial filter OptBasisGNN (Guo & Wei, 2023) orthogonalizes the polynomial basis in order to maximize the convergence speed. Instead of using fixed-order polynomials, Specformer (Bo et al., 2023) resorts to Transformer (Vaswani et al., 2017) to derive learnable bases for each feature dimension. While Specformer demonstrates promising performance, it requires conducting eigendecomposition with the cost of $O(n^3)$, rendering it impractical for large social graphs. Contrarily, the time complexity of UniFilter is *linear* to both graph sizes and propagation hops. Nonetheless, none of the above polynomial filters take the varying heterophily degrees of graphs into consideration when utilizing polynomials, which leads to suboptimal empirical performance, as verified in our experiments.

**Model-optimized GNNs.** One commonly adopted technique in model design is to combine both low-pass and high-pass filters. GNN-LF/HF (Zhu et al., 2021) devises variants of the Laplacian matrix to construct a low-pass and high-pass filter respectively. HOG-GNN Wang et al. (2022) designs a new propagation mechanism and considers the heterophily degrees between node pairs during neighbor aggregation, which is optimized from a spatial perspective. DualGR Ling et al. (2023) focuses on the multi-view graph clustering problem and proposes dual label-guided graph refinement to handle heterophily graphs, which is a graph-level classification task. ACM-GCN (Luan et al., 2022) trains both low-pass and high-pass filters in each layer and then adopts the embeddings from each filter adaptively. Another applied model design aims to extract homophily from both local and global graph structures. $H_2$GCN (Zhu et al., 2020) incorporates ego and neighbor embeddings, and high-order neighborhood and intermediate representations. Similarly, GloGNN++ (Li et al., 2022) trains a coefficient matrix in each layer to measure the correlations between nodes so as to aggregate homophilous nodes globally. To explicitly capture the relations between distant nodes, WRGAT (Suresh et al., 2021) leverages the graph rewiring (Topping et al., 2022; Karhadkar et al., 2023) technique by constructing new edges with weights to measure node proximity. Additionally, there are GNNs handling heterophily graphs from other aspects. LINKX (Lim et al., 2021) learns embeddings from node features and graph structure in parallel. Then the two embeddings are concatenated and fed into MLP for node predictions. Orderred GNN (Song et al., 2023) establishes the hierarchy structure of neighbors and then constrains the neighbor nodes within specific hops into the specific blocks of neurons, avoiding feature mixing within hops.

## 7 CONCLUSION

In this paper, we propose a universal polynomial basis UniBasis by incorporating the graph heterophily degrees in the premise of thorough theoretical analysis for spectral graph neural networks. Upon UniBasis, we devise a general graph filter UniFilter. By a comprehensive evaluation of UniFilter on both real-world and synthetic datasets against a wide range of baselines, the remarkably superior performance of UniFilter significantly supports the effectiveness and generality of UniBasis for graphs with varying heterophily. In addition, UniBasis is exhibited as a promising new method for graph analysis by capturing the spectral characteristics of graphs and enriching the interpretability.

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

# A APPENDIX

## A.1 PROOFS

**Proof 1 (Proof of Lemma 2.1)** *Given any normalized signal* $\mathbf{x} \in \mathbb{R}^n$ *on graph* $\mathbf{G} = (\mathcal{V}, \mathcal{E})$, $f(\mathbf{x}) = \frac{\mathbf{x}^\top \mathbf{L} \mathbf{x}}{2} = \frac{\sum_{\langle u,v \rangle \in \mathcal{E}} (\mathbf{x}_u - \mathbf{x}_v)^2}{2 \sum_{u \in \mathcal{V}} \mathbf{x}_u^2 d_u}$. *It is straightforward to infer* $f(\mathbf{x}) \geq 0$. *Meanwhile, it is known that* $\sup_{\mathbf{x} \in \mathbb{R}^n} \frac{\sum_{\langle u,v \rangle \in \mathcal{E}} (\mathbf{x}_u - \mathbf{x}_v)^2}{\sum_{u \in \mathcal{V}} \mathbf{x}_u^2 d_u} \leq 2$ *(Chung & Graham, 1997). Therefore* $f(\mathbf{x}) \leq 1$, *which completes the proof.*

**Proof 2 (Proof of Theorem 1)** *Let* $\mathbf{z} = \mathrm{F}(\mathbf{w})\mathbf{x} = \sum_{k=0}^{K} \mathbf{w}_k \mathbf{P}^k \mathbf{x}$. *Then frequency* $f(\sum_{k=0}^{K} \mathbf{w}_k \mathbf{P}^k \mathbf{x}) = f(\mathbf{z}) = \frac{\mathbf{z}^\top \mathbf{L} \mathbf{z}}{2} = \frac{\sum_{\langle u,v \rangle \in \mathcal{E}} (\mathbf{z}_u - \mathbf{z}_v)^2}{2 \sum_{u \in \mathcal{V}} \mathbf{z}_u^2 d_u}$. *It is noteworthy that* $\mathrm{F}(\mathbf{w})$ *represents the optimal filter for node classification. This implies that node representations acquired by* $\mathrm{F}(\mathbf{w})$ *exhibit similarity among nodes belonging to the same classes and dissimilarity among nodes belonging to distinct classes.*

*W.l.o.g, for* $\forall u, v \in \mathcal{V}$, *we assume a constant* $\delta$ *that* $|\mathbf{z}_u - \mathbf{z}_v| \leq c\delta$ *with* $c \ll 1$ *if* $\mathbf{Y}_u = \mathbf{Y}_v$; *otherwise* $|\mathbf{z}_u - \mathbf{z}_v| = g(\mathbf{Y}_u, \mathbf{Y}_v)\delta$ *where* $\mathbf{Y} \in \mathbb{N}^{n \times |\mathcal{C}|}$ *is the one-hot label matrix and* $g(\mathbf{Y}_u, \mathbf{Y}_v) \geq 1$ *is a constant determined by* $\mathbf{Y}_u$ *and* $\mathbf{Y}_v$. *As a result, we have* $\frac{\sum_{\langle u,v \rangle \in \mathcal{E}} (\mathbf{z}_u - \mathbf{z}_v)^2}{2 \sum_{u \in \mathcal{V}} \mathbf{z}_u^2 d_u}$ *approaches to* $\frac{c^2 \delta^2 hm + \sum_{\langle u,v \rangle \in \mathcal{E}, \mathbf{Y}_u \neq \mathbf{Y}_v} g^2(\mathbf{Y}_u, \mathbf{Y}_v)\delta^2}{2 \sum_{u \in \mathcal{V}} \mathbf{z}_u^2 d_u}$. *Since* $c \ll 1$ *and* $g(\mathbf{Y}_u, \mathbf{Y}_v) \geq 1$, *therefore* $g^2(\mathbf{Y}_u, \mathbf{Y}_v) \gg c^2$ *holds. In this regard, frequency* $f(\sum_{k=0}^{K} \mathbf{w}_k \mathbf{P}^k \mathbf{x})$ *is dominated by* $\frac{\sum_{\langle u,v \rangle \in \mathcal{E}, \mathbf{Y}_u \neq \mathbf{Y}_v} g^2(\mathbf{Y}_u, \mathbf{Y}_v)\delta^2}{2 \sum_{u \in \mathcal{V}} \mathbf{z}_u^2 d_u}$, *i.e., proportional to* $1 - h$.

**Proof 3 (Proof of Theorem 2)** *Let* $\{\lambda_1, \lambda_2, \cdots, \lambda_n\}$ *be the eigenvalues of* $\mathbf{P}$ *associated with eigenvectors* $\{\mathbf{v}_1, \mathbf{v}_2, \cdots, \mathbf{v}_n\}$. *For a general (non-bipartite) connected graph* $\mathbf{G}$, *we have* $-1 < \lambda_1 \leq \lambda_2 \leq \cdots \leq \lambda_n = 1$ *and* $\mathbf{v}_i^\top \mathbf{v}_j = 0$ *for* $i \neq j$ *and* $\mathbf{v}_i^\top \mathbf{v}_j = 1$ *for* $i = j$. *In particular, we have* $\lambda_n = 1$ *and* $\mathbf{v}_n = \frac{\mathbf{D}^{\frac{1}{2}}\mathbf{1}}{\sqrt{2m}}$ *where* $\mathbf{1} \in \mathbb{R}^n$ *is the all-one vector. Consequently, we have* $\mathbf{P}^k \mathbf{x} = \sum_{i=1}^{n} \lambda_i^k (\mathbf{v}_i^\top \mathbf{x})\mathbf{v}_i$ *and* $\|\mathbf{P}^k \mathbf{x}\| = \sqrt{\sum_{i=1}^{n} (\lambda_i^k \mathbf{v}_i^\top \mathbf{x})^2}$. *Hence,* $\mathbf{P}^k \mathbf{x} \cdot \mathbf{P}^{k+1} \mathbf{x} = \sum_{i=1}^{n} \lambda_i^k (\mathbf{v}_i^\top \mathbf{x}) \cdot \lambda_i^{k+1} (\mathbf{v}_i^\top \mathbf{x}) = \sum_{i=1}^{n} \lambda_i^{2k+1} (\mathbf{v}_i^\top \mathbf{x})^2$. *For ease of exposition, we set constants* $c_1 = \|\mathbf{P}^k \mathbf{x}\|$ *and* $c_2 = \|\mathbf{P}^{k+1} \mathbf{x}\|$. *Therefore,* $\frac{\mathbf{P}^k \mathbf{x} \cdot \mathbf{P}^{k+1} \mathbf{x}}{\|\mathbf{P}^k \mathbf{x}\| \|\mathbf{P}^{k+1} \mathbf{x}\|} = \sum_{i=1}^{n} \frac{\lambda_i^k (\mathbf{v}_i^\top \mathbf{x})}{c_1} \cdot \frac{\lambda_i^{k+1} (\mathbf{v}_i^\top \mathbf{x})}{c_2} = \sum_{i=1}^{n} \frac{\lambda_i^{2k+1} (\mathbf{v}_i^\top \mathbf{x})^2}{c_1 c_2}$. *W.l.o.g, let* $t$ *be the integer such that* $\lambda_t < 0 \leq \lambda_{t+1}$. *Notice that* $\sum_{i=1}^{n} \left( \frac{\lambda_i^k (\mathbf{v}_i^\top \mathbf{x})}{c_1} \right)^2 = \sum_{i=1}^{n} \left( \frac{\lambda_i^{k+1} (\mathbf{v}_i^\top \mathbf{x})}{c_1} \right)^2 = 1$ *hold for all* $k$ *and* $c_1$ *and* $c_2$ *are normalization factors, exerting equal scaling effects to both the negative part and the positive part. As a result, the negative value* $\sum_{i=1}^{t} \frac{\lambda_i^{2k+1} (\mathbf{v}_i^\top \mathbf{x})^2}{c_1 c_2}$ *with* $\lambda_i \in (-1, 0)$ *decreases and the positive value* $\sum_{i=t}^{n} \frac{\lambda_i^{2k+1} (\mathbf{v}_i^\top \mathbf{x})^2}{c_1 c_2}$ *with* $\lambda_i \in (0, 1]$ *increases as the exponent* $k$ *increases. As a consequence,* $\frac{\mathbf{P}^k \mathbf{x} \cdot \mathbf{P}^{k+1} \mathbf{x}}{\|\mathbf{P}^k \mathbf{x}\| \|\mathbf{P}^{k+1} \mathbf{x}\|}$ *is monotonically increasing with* $k$, *and thus the angle angle* $\arccos \left( \frac{\mathbf{P}^k \mathbf{x} \cdot \mathbf{P}^{k+1} \mathbf{x}}{\|\mathbf{P}^k \mathbf{x}\| \|\mathbf{P}^{k+1} \mathbf{x}\|} \right)$ *is progressively smaller.*

*Similarly, we have* $\mathbf{P}^K \mathbf{x} \cdot \psi = \sum_{i=1}^{n} \lambda_i^K (\mathbf{v}_i^\top \mathbf{x})\mathbf{v}_i \cdot \psi$. *Since* $\lambda_n = 1$ *and* $\mathbf{v}_n = \frac{\mathbf{D}^{\frac{1}{2}}\mathbf{1}}{\sqrt{2m}}$, *we have* $\lim_{K \to \infty} \mathbf{P}^K \mathbf{x} = \lambda_n^K (\mathbf{v}_n^\top \mathbf{x})\mathbf{v}_n = \frac{\mathbf{v}_n^\top \mathbf{x}}{\sqrt{2m}} \mathbf{D}^{\frac{1}{2}} \mathbf{1}$ *which is scalar multiple of* $\psi$. *Therefore,* $\lim_{K \to \infty} \arccos \left( \frac{\mathbf{P}^K \mathbf{x} \cdot \psi}{\|\mathbf{P}^K \mathbf{x}\| \|\psi\|} \right) = 0$ *holds.*

**Proof 4 (Proof of Theorem 3)** *W.l.o.g, we consider a* $k$-*regular graph* $\mathbf{G} = (\mathcal{V}, \mathcal{E})$ *with* $n$ *nodes. Given a random normalized signal* $\mathbf{x} = (\mathbf{x}_1, \mathbf{x}_2, \cdots, \mathbf{x}_n)^\top$, $\phi \cdot \mathbf{x} = \sum_{i=1}^{n} \frac{\mathbf{x}_i}{\sqrt{n}}$. *Meanwhile,* $f(\mathbf{x}) = \frac{\mathbf{x}^\top \mathbf{L} \mathbf{x}}{2} = \frac{\sum_{\langle u,v \rangle \in \mathcal{E}} (\mathbf{x}_u - \mathbf{x}_v)^2}{2 \sum_{u \in \mathcal{V}} \mathbf{x}_u^2 d_u} = \frac{\sum_{\langle u,v \rangle \in \mathcal{E}} (\mathbf{x}_u - \mathbf{x}_v)^2}{2k}$. *Over the randomness of* $\mathbf{G}$, *the expectation of the*

*spectral signal frequency*

$$\mathbb{E}_{\mathbf{G} \sim \mathcal{G}}[f(\mathbf{x})] = \frac{1}{2k} \cdot \frac{k}{n-1} \sum_{\forall u,v \in \mathcal{V}} \frac{(\mathbf{x}_u - \mathbf{x}_v)^2}{2}$$

$$= \frac{1}{4(n-1)} \left( \sum_{u \in \mathcal{V}} (n-1)\mathbf{x}_u^2 - \sum_{u \in \mathcal{V}} 2\mathbf{x}_u \big( \sum_{v \in \mathcal{V} \setminus \{u\}} \mathbf{x}_v \big) \right)$$

$$= \frac{1}{4(n-1)} \left( n - 1 - \sum_{u \in \mathcal{V}} 2\mathbf{x}_u \big( \sum_{i=1}^{n} \mathbf{x}_i - \mathbf{x}_u \big) \right)$$

$$= \frac{1}{4(n-1)} \left( n - 1 - 2\big( \sum_{i=1}^{n} \mathbf{x}_i \big)^2 + 2 \right)$$

$$= \frac{1}{4(n-1)} \left( n + 1 - 2\big( \sum_{i=1}^{n} \mathbf{x}_i \big)^2 \right)$$

$$= \frac{n+1}{4(n-1)} - \frac{1}{2(n-1)} \big( \sum_{i=1}^{n} \frac{\mathbf{x}_i}{\sqrt{n}} \big)^2$$

$$= \frac{n+1-2(\phi \cdot \mathbf{x})^2}{4(n-1)}$$

*As a consequence, if the angle $\theta := \arccos(\phi \cdot \mathbf{x})$ increases, $\phi \cdot \mathbf{x}$ decreases, resulting the increment of $\mathbb{E}_{\mathbf{G} \sim \mathcal{G}}[f(\mathbf{x})]$, which completes the proof.*

Before the proof of Theorem 4, we first introduce the following Lemma.

**Lemma A.1 (Proposition 4.3 (Guo & Wei, 2023))** *Vector $\mathbf{v}_k$ in Algorithm 1 is only dependent with $\mathbf{v}_{k-1}$ and $\mathbf{v}_{k-2}$.*

**Proof 5 (Proof of Theorem 4)** *Based on Lemma A.1, it is easy to prove that $\{\mathbf{v}_0, \mathbf{v}_1, \cdots, \mathbf{v}_K\}$ forms an orthonormal basis. Before proceeding, we first demonstrate that $\mathbf{v}_{k+1}$ is orthogonal to $\{\mathbf{u}_0, \mathbf{u}_1, \cdots, \mathbf{u}_k\}$ for $k \in \{0, 1, \cdots, K-1\}$. In particular, we have $\mathbf{u}_0 = \mathbf{v}_0$ and $\mathbf{u}_k = \frac{\mathbf{s}_{k-1}/k + t_k \mathbf{v}_k}{\|\mathbf{s}_{k-1}/k + t_k \mathbf{v}_k\|} = \frac{\frac{1}{k}\sum_{i=0}^{k-1} \mathbf{u}_i + t_k \mathbf{v}_k}{\|\frac{1}{k}\sum_{i=0}^{k-1} \mathbf{u}_i + t_k \mathbf{v}_k\|}$ for $k \in [K]$ according to Algorithm 1. Therefore, it is intuitive that there exist constants $\alpha_0, \alpha_1, \cdots, \alpha_k$ such that $\mathbf{u}_k = \sum_{i=0}^{k} \alpha_i \mathbf{v}_i$ holds. Since $\{\mathbf{v}_0, \mathbf{v}_1, \cdots, \mathbf{v}_K\}$ is an orthonormal basis, therefore $\mathbf{v}_{k+1}$ is orthogonal to $\{\mathbf{u}_0, \mathbf{u}_1, \cdots, \mathbf{u}_k\}$.*

*Denote $\theta := \frac{(1-h)\pi}{2}$. First, we prove that $\mathbf{u}_0 \cdot \mathbf{u}_1 = \cos(\theta)$. In particular, we have $\mathbf{u}_1 = \frac{\mathbf{u}_0 + t_1 \mathbf{v}_1}{\|\mathbf{u}_0 + t_1 \mathbf{v}_1\|}$ and $t_1 = \sqrt{(\frac{\mathbf{s}_0^\top \mathbf{u}_0}{\cos(\theta)})^2 - 1} = \sqrt{\frac{1}{\cos^2(\theta)} - 1}$. Then $\|\mathbf{u}_0 + t_1 \mathbf{v}_1\| = \sqrt{1 + t_1^2} = \frac{1}{\cos(\theta)}$. Hence, $\mathbf{u}_0 \cdot \mathbf{u}_1 = \mathbf{u}_0^\top (\mathbf{u}_0 + t_1 \mathbf{v}_1)\cos(\theta) = \cos(\theta)$.*

*Second, we assume that $\mathbf{u}_i \cdot \mathbf{u}_j = \cos(\theta)$ holds for $\forall i, j \in \{0, 1, \cdots, k-1\}$ and $i \neq j$. In what follows, we then prove that $\mathbf{u}_k \cdot \mathbf{u}_j = \cos(\theta)$ holds for $j \in \{0, 1, \cdots, k-1\}$. Specifically, we have $\mathbf{u}_k = \frac{\frac{1}{k}\sum_{i=0}^{k-1} \mathbf{u}_i + t_k \mathbf{v}_k}{\|\frac{1}{k}\sum_{i=0}^{k-1} \mathbf{u}_i + t_k \mathbf{v}_k\|}$. In particular, for the denominator, we have*

$$\left\| \frac{1}{k} \sum_{i=0}^{k-1} \mathbf{u}_i + t_k \mathbf{v}_k \right\| = \sqrt{ \left( \frac{1}{k} \sum_{i=0}^{k-1} \mathbf{u}_i^\top + t_k \mathbf{v}_k^\top \right) \left( \frac{1}{k} \sum_{i=0}^{k-1} \mathbf{u}_i + t_k \mathbf{v}_k \right) }$$

$$= \sqrt{ \frac{\sum_{i=0}^{k-1} \mathbf{u}_i^\top \mathbf{u}_i + 2 \sum_{i=0}^{k-2} \mathbf{u}_i^\top (\sum_{j=i+1}^{k-1} \mathbf{u}_j)}{k^2} + t_k^2 }$$

$$= \sqrt{ \frac{k + k(k-1)\cos(\theta)}{k^2} + \left( \frac{\mathbf{s}_{k-1}^\top \mathbf{u}_{k-1}}{k\cos(\theta)} \right)^2 - \frac{(k-1)\cos(\theta)+1}{k} }$$

$$= \frac{\sum_{i=0}^{k-1} \mathbf{u}_i^\top \cdot \mathbf{u}_{k-1}}{k\cos(\theta)}$$

$$= \frac{1 + (k-1)\cos(\theta)}{k\cos(\theta)}$$

**Table 4: Dataset Details.**

| Dataset | Cora | Citeseer | Pubmed | Actor | Chameleon | Squirrel |
|---|---|---|---|---|---|---|
| #Nodes ($n$) | 2,708 | 3,327 | 19,717 | 7,600 | 2,277 | 5,201 |
| #Edges ($m$) | 5,429 | 4,732 | 44,338 | 26,659 | 31,371 | 198,353 |
| #Features ($d$) | 1,433 | 3,703 | 500 | 932 | 2,325 | 2,089 |
| #Classes | 7 | 6 | 3 | 5 | 5 | 5 |
| Homo. ratio ($h$) | 0.81 | 0.73 | 0.80 | 0.22 | 0.23 | 0.22 |

**Table 5: $\tau$ Selection.**

| Dataset | Cora | Citeseer | Pubmed | Actor | Chameleon | Squirrel |
|---|---|---|---|---|---|---|
| $\tau$ | 1.0 | 0.9 | 0.8 | 0.1 | 0.7 | 0.7 |

*Meanwhile, we have* $\mathbf{u}_k \cdot \mathbf{u}_j = \frac{\frac{1}{k}\sum_{i=0}^{k-1}\mathbf{u}_i^\top \mathbf{u}_j + t_k \mathbf{v}_k^\top \mathbf{u}_j}{\|\frac{1}{k}\sum_{i=0}^{k-1}\mathbf{u}_i + t_k \mathbf{v}_k\|} = \frac{\frac{1}{k}(1+(k-1)\cos(\theta))\cdot k\cos(\theta)}{1+(k-1)\cos(\theta)} = \cos(\theta).$

*Eventually, it is easy to verify that the derivation holds for* $\forall j \in \{0, 1, \cdots, k-1\}$*, which completes the proof.*

### A.2 EXPERIMENTAL SETTINGS

**Datasets.** Table 4 presents the detailed statistics of the six real-world datasets. The three homophily datasets, i.e.,*Cora*, *Citeseer*, and *Pubmed* are citation networks. Each graph node represents a research paper, and each edge denotes a citation relationship. Feature vectors of nodes are bag-of-words representations. The one-hot label assigned to each node stands for one research field of the paper. The rest three datasets are heterophily datasets. Specifically, *Actor* is a co-occurrence graph from the film-director-actor-writer network from WebKB3 (Tang et al., 2009; Pei et al., 2020). *Squirrel* and *Chameleon* are two datasets extracted from Wikipedia web pages, and nodes are categorized by the average amounts of monthly traffic (Li et al., 2022).

while the three heterophily datasets are Wikipedia datasets and WebKB3 dataset with homophily ratios around 0.22.

**Running environment.** All our experiments are conducted on a Linux machine with an RTX2080 GPU (10.76GB memory), Xeon CPU, and 377GB RAM.

**Parameter settings.** During the training process, learnable parameters are tuned with Adam (Kingma & Ba, 2015) optimizer. We set a patience of early stopping with 200 epochs. For hyperparameters, we fix propagation hop $K = 10$ for all tested datasets. The rest hyperparameters are selected in the following ranges.

1. Learning rate: $[0.001, 0.005, 0.01, 0.05, 0.1, 0.15, 0.2]$;

2. Hidden dimension: $[64, 128, 256]$;

3. MLP layers: $[2, 3, 4]$;

4. Weight decays: $[0, 1e-4, 5e-4, 0.001]$;

5. Drop rates: $[0, 0.1, 0.2, \cdots, 0.9]$.

**Choice of $\tau$.** Ideally, the selection of $\tau$ for each dataset is highly related to its homophily ratio. Normally, we tune $\tau$ near the range of $h \pm 0.5$. In our experiments, the $\tau$ values are set as in Table 5.

### A.3 ADDITIONAL EXPERIMENTS

**Homophily ratio estimation.** For each dataset, we generate 10 random splits of training/validation/test are generated. For each split, we estimate the homophily ratio from the training data, using it as the input to UniFilter to construct polynomial bases for node classification. Notice that we estimate a new homophily ratio for each split and then obtain the corresponding ac-

**Table 6: Estimated homophily ratios $\hat{h}$.**

| Dataset | Cora | Citeseer | Pubmed | Actor | Chameleon | Squirrel |
|---|---|---|---|---|---|---|
| $\hat{h}_1$ | $0.82 \pm 0.01$ | $0.70 \pm 0.01$ | $0.79 \pm 0.005$ | $0.21 \pm 0.004$ | $0.24 \pm 0.01$ | $0.22 \pm 0.005$ |
| $\hat{h}_2$ | $0.82 \pm 0.01$ | $0.69 \pm 0.014$ | $0.79 \pm 0.01$ | $0.21 \pm 0.004$ | $0.24 \pm 0.01$ | $0.22 \pm 0.01$ |

**Table 7: Estimated homophily ratio $\hat{h}$ over varying training percentages on Cora and Squirrel.**

| Dataset | 10% | 20% | 30% | 40% | 50% | 60% |
|---|---|---|---|---|---|---|
| Cora | $0.83 \pm 0.05$ | $0.83 \pm 0.04$ | $0.83 \pm 0.03$ | $0.83 \pm 0.01$ | $0.82 \pm 0.08$ | $0.82 \pm 0.01$ |
| Squirrel | $0.23 \pm 0.014$ | $0.22 \pm 0.011$ | $0.22 \pm 0.010$ | $0.22 \pm 0.006$ | $0.22 \pm 0.005$ | $0.22 \pm 0.005$ |

**Table 8: Estimated homophily ratio $\hat{h}$ over varying training percentages on Cora and Squirrel.**

| Varying $\hat{h}$ | 0.78 | 0.79 | 0.80 | 0.81 | 0.82 |
|---|---|---|---|---|---|
| Squirrel | $91.03 \pm 0.61$ | $91.28 \pm 0.64$ | $91.34 \pm 0.62$ | $91.19 \pm 0.67$ | $91.17 \pm 0.69$ |
| **Varying $\hat{h}$** | 0.19 | 0.20 | 0.21 | 0.22 | 0.23 |
| Pubmed | $66.22 \pm 1.43$ | $66.96 \pm 1.38$ | $67.09 \pm 1.08$ | $67.01 \pm 1.25$ | $66.69 \pm 1.26$ |

curacy score. We present the estimated homophily ratio $\hat{h}$ with standard deviation averaged over the 10 training sets for all datasets in Table 6. In particular, we denote the averaged homophily ratio estimations as $\hat{h}_1$ and $\hat{h}_2$ for data splits settings of $60\%/20\%/20\%$ for polynomial filters and $48\%/32\%/20\%$ for model-optimized methods respectively. To verify the estimation difficulty, we vary the percentages of the training set in $\{10\%, 20\%, 30\%, 40\%, 50\%, 60\%\}$ on Cora and Squirrel, and then average the estimated homophily ratios $\hat{h}$ over 10 random splits. The results are presented in Table 7.

As shown in Table 6, the estimated values $\hat{h}_1$ and $\hat{h}_2$ derived from training datasets closely align with the actual homophily ratio $h$. There is a difference within a $2\%$ range across all datasets, except for Citeseer. As shown in Table 6, the estimated homophily ratio $\hat{h}$ is approaching the true homophily ratio $h$ across varying percentages of training data. This observation verifies that a high-quality estimation of the homophily ratio is accessible by the training set.

**Sensitivity of UniFilter to $\hat{h}$.** We conduct an ablation study on the sensitivity of our proposed UniFilter to the estimated homophily ratio. Since UniFilter only utilizes the homophily basis on Cora ($\tau = 1$ for Cora as shown in Table 5 in the Appendix in submission) and does not rely on the homophily ratio, we thus test UniFilter on Pubmed and Squirrel. To this end, we vary the estimated homophily ratio $\hat{h} \in \{0.78, 0.79, 0.80, 0.81, 0.82\}$ on Pubmed and $\hat{h} \in \{0.19, 0.20, 0.21, 0.22, 0.23\}$ on Squirrel based on the results in Table 7. The achieved accuracy scores are presented in Table 8.

In Table 8, UniFilter demonstrates consistent accuracy scores across different homophily ratios for both Pubmed and Squirrel. Notably, the accuracy variations remain minor, staying within a $1\%$ range from the scores under the true homophily ratio, particularly on the Pubmed dataset.

**Mitigation of Over-smoothing.** As claimed, our proposed method UniFilter could address the over-smoothing problem. To verify in experiments, we generate a 1000-length of homophily basis and our proposed heterophily basis on dataset Squirrel respectively, i.e., consisting of 1000 basis vectors. We calculate the degrees of the angle formed by all two consecutive basis vectors and present the degree distribution in Table 9. Notice that the degree is averaged across the dimension of node features.

As shown, degrees of the formed angles by two consecutive basis vectors from the homophily basis approach to $0°$, which indicates that the homophily basis on Squirrel converges asymptotically. On the contrary, degrees of our basis keep around $71°$ determined by our setting $\theta = \frac{\pi}{2}(1 - h)$ where

Table 9: Degree ($°$) distribution of the homophily basis and UniBasis on Squirrel ($h = 0.22$).

| Basis | $(v_1, v_2)$ | $(v_2, v_3)$ | $(v_3, v_4)$ | $(v_4, v_5)$ | $\cdots$ | $(v_{996}, v_{997})$ | $(v_{997}, v_{998})$ | $(v_{998}, v_{999})$ | $(v_{999}, v_{1000})$ |
|---|---|---|---|---|---|---|---|---|---|
| Homo. Basis | $88.74°$ | $87.99°$ | $87.76°$ | $86.51°$ | $\cdots$ | $0.0123°$ | $0.0118°$ | $0.0115°$ | $0.0114°$ |
| UniBasis | $69.72°$ | $70.03°$ | $70.01°$ | $70.05°$ | $\cdots$ | $71.37°$ | $71.23°$ | $71.32°$ | $71.22°$ |

$h = 0.22$ for Squirrel. Notice that the variation of degrees is due to the average across 2089-dimension features of Squirrel.

