# OpenReview forum: "An Effective Universal Polynomial Basis for Spectral Graph Neural Networks"
_ICLR.cc/2024/Conference — Submitted to ICLR 2024_

### Official Review · Reviewer_bWo5 · 2023-10-30

**Soundness:** 2 fair
**Presentation:** 2 fair
**Contribution:** 2 fair
**Rating:** 3
**Confidence:** 5

**Summary:**

The paper introduces UniBasis, a universal polynomial basis designed to align with varying degrees of graph heterophily. UniBasis is used to create UniFilter, a general graph filter. The authors demonstrate that UniFilter outperforms 18 baseline methods on real-world and synthetic datasets, confirming UniBasis's effectiveness and generality for graph analysis, particularly on graphs with varying levels of heterophily.

**Strengths:**

1) Taking into account the diverse degrees of heterophily when designing polynomial bases is an interesting idea that holds the potential to enhance filter learning.
2) The proposed heterophily bases demonstrate both theoretical and practical soundness, proving effective in somewhat.

**Weaknesses:**

1) The design of heterophilic bases relies on the dataset's homophily rate, denoted as $h$ in Algorithm 1. I am concerned this approach is impractical due to obtaining the exact homophily rate $h$ from the training data is not feasible. It appears that the authors have directly utilized the entire dataset, including the labels of the test set. There are also methods to learn the homophily rate $h$ during the training process,  but I think this process might affect the model's performance.
2) There are not enough datasets for heterophilic graphs, and previous work has highlighted some issues with the Chameleon and Squirrel datasets [1]. Therefore, I recommend conducting experiments using more extensive heterophilic graph datasets, such as the latest benchmarks available [1] and [2].

Minor Comments:
1) The writing in this paper requires further refinement. For example, the notations used are somewhat confusing, such as using bold uppercase letters $\mathbf{G}$ for a graph and calligraphic fonts $\mathcal{L}$ for the Laplacian matrix.

2) No available code for reproducing the results has been provided.

[1] Platonov, Oleg, et al. "A critical look at the evaluation of GNNs under heterophily: Are we really making progress?." _The Eleventh International Conference on Learning Representations_. 2022.

[2] Lim, Derek, et al. "Large scale learning on non-homophilous graphs: New benchmarks and strong simple methods." _Advances in Neural Information Processing Systems_ 34 (2021): 20887-20902.

**Questions:**

1) Please refer to the aforementioned weaknesses.
2) I don't have any more concerns. My main concern is for the use of $h$, which brings some unfairness, and I would like to see a further response from the author.

---

> ### Author Response · Authors · 2023-11-18
> **Response to Reviewer bWo5**
>
> Dear Reviewer bWo5,
>
> We sincerely thank you for your valuable feedback on our paper. We address your comments and questions below. In the revised draft, we have updated the paper according to the reviewer's points. We mark our revision in $\textcolor{blue}{blue}$.
>
>
>
> **W1**: The design of heterophilic bases relies on the dataset's homophily rate ... but I think this process might affect the model's performance.
>
> **A1**: Please kindly refer to our global response. As demonstrated, a high-quality estimation of the homophily ratio is accessible by the training set without compromising the performance of our model. As advised, we have added a discussion related to the homophily ratio and updated the experimental results in our revision.
>
> Additionally, we have conducted an experiment to investigate the sensitivity of our proposed UniFilter to the estimated homophily ratio. Since UniFilter only utilizes the homophily basis on Cora ($\tau=1$ for Cora as shown in Table 5 in the Appendix in submission) and does not rely on the homophily ratio, we thus test UniFilter on Pubmed and Squirrel. To this end, we vary the estimated homophily ratio $\hat{h} \in \\{0.78, 0.79, 0.80, 0.81, 0.82\\}$ on Pubmed and  $\hat{h} \in\\{0.19,0.20,0.21,0.22,0.23\\}$ on Squirrel based on the results in Table C (in global response). The achieved accuracy scores are presented in Table H.
>
> **Table H: Accuracy (%) for varying estimated homophily ratio $\hat{h}$ on Pubmed and Squirrel.**
>
> | Varying $\hat{h}$     | 0.78             | 0.79             | 0.80 (true $h$)     | 0.81             | 0.82             |
> | --------------------- | ---------------- | ---------------- | ------------------- | ---------------- | ---------------- |
> | Pubmed                | 91.03 $\pm$ 0.61 | 91.28 $\pm$ 0.64 | 91.34 $\pm$ 0.62    | 91.19 $\pm$ 0.67 | 91.17 $\pm$ 0.69 |
> | **Varying $\hat{h}$** | **0.19**         | **0.20**         | **0.21 (true $h$)** | **0.22**         | **0.23**         |
> | Squirrel              | 66.22 $\pm$ 1.43 | 66.96 $\pm$ 1.38 | 67.09 $\pm$ 1.08    | 67.01 $\pm$ 1.25 | 66.69 $\pm$ 1.26 |
>
> In Table H, UniFilter demonstrates consistent accuracy scores across different homophily ratios for both Pubmed and Squirrel. Notably, the accuracy variations remain minor, staying within a 1\% range from the scores under the true homophily ratio, particularly on the Pubmed dataset.
>
> Based on the experimental results, we observe that i) the homophily ratio can be effectively estimated using training data, and ii) our proposed UniFilter is robust towards the variance of estimated homophily ratios.
>
>
>
> **W2**: There are not enough datasets for heterophilic graphs ... more extensive heterophilic graph datasets, such as the latest benchmarks available [1] and [2].
>
> **A2**: As advised, we have tested our model on the new heterophily dataset amazon-ratings from [1]. Our UniFilter is tested on amazon-ratings by following the settings in [1] in a fixed dataset splits of 50\%/25\%/25\% for training/validation/test for a quick comparison.
>
> Specifically, UniFilter achieves an accuracy score of 53.28\% $\pm$ 0.71\% on amazon-ratings, slightly lower than the highest score 53.63\% $\pm$ 0.39\% achieved by GaphSage, as reported in [1]. However, UniFilter beats all heterophily-specific models tested in [1]. This observation confirms the effectiveness of our method as a spectral filter.
>
> [1] Platonov, Oleg, et al. "A critical look at the evaluation of GNNs under heterophily: Are we really making progress?." *The Eleventh International Conference on Learning Representations*. 2022.
>
> [2] Lim, Derek, et al. "Large scale learning on non-homophilous graphs: New benchmarks and strong simple methods." *Advances in Neural Information Processing Systems* 34 (2021): 20887-20902.
>
>
>
> Minor Comments:
>
> **C1**: The writing in this paper requires further refinement. For example, the notations used are somewhat confusing, such as using bold uppercase letters $\mathbf{G}$ for a graph and calligraphic fonts $\mathcal{L}$ for the Laplacian matrix.
>
> **A1**: We thank the reviewer for the comments. As advised, we have replaced the calligraphic fonts $\mathcal{L}$ with bold uppercase letter $\mathbf{L}$ in our revision.
>
> **C2**: No available code for reproducing the results has been provided.
>
> **A2**: As advised, we have uploaded the code with instructions to reproduce our experimental results.
>
>
>
> Thank you again for your valuable time and effort spent reviewing.

---

> > ### Comment · Reviewer_bWo5 · 2023-11-20
> > **Response**
> >
> > Thank you for the prompt response from the authors. Using the training set to estimate the homophily ratio is a solution, but is it still feasible when the training set is smaller? For example, the standard splitting in GCN, i.e., 20 nodes per class for training, and the sparse splitting used in GPRGNN and ChebNetII, i.e., 2.5% of nodes for training.
> >
> > More heterophilic graphs are still needed for experiments. Please note that the Chameleon and Squirrel datasets are controversial.

---

> ### Author Response · Authors · 2023-11-21
> **Further response**
>
> We thank the reviewer for the replies.
>
> As suggested, we estimate the homophily ratio when only utilizing 2.5% of nodes for training. Table I below presents the estimated homophily ratio $\hat{h}$ for all the tested datasets.
>
>
>
> **Table I: Estimated homophily ratio $\hat{h}$ using 2.5% of nodes for training**
>
> | Dataset             | Cora             | Citeseer         | Pubmed           | Actor            | Chameleon        | Squirrel         |
> | ------------------- | ---------------- | ---------------- | ---------------- | ---------------- | ---------------- | ---------------- |
> | True $h$            | 0.81             | 0.73             | 0.80             | 0.22             | 0.23             | 0.22             |
> | Estimated $\hat{h}$ | 0.847 $\pm$ 0.22 | 0.880 $\pm$ 0.21 | 0.784 $\pm$ 0.06 | 0.203 $\pm$ 0.09 | 0.246 $\pm$ 0.07 | 0.254 $\pm$ 0.05 |
>
> The estimated homophily ratio $\hat{h}$ is close to the true $h$ across all datasets except for Citeseer. By further testing, we find that it requires 6% nodes as training for an accurate estimation of $0.732 \pm 0.12$ on Citeseer. In such a case for the sparse setting, we can take the homophily ratio as a hyperparameter and tune it within the vicinity of the estimated $\hat{h}$ until we achieve the best possible performance.
>
> As required, we test our model UniFilter on one new heterophily dataset Texas from the suggested ChebNetII paper (due to time constraints, only one new dataset is considered). By following the full-supervised setting, our model UniFilter achieves an accuracy score of **93.44% $\pm$ 3.59%** on Texas, higher than the best result of **93.28% $\pm$ 1.47%** presented in Table 6 in ChebNetII paper. We believe this result provides further confirmation of the effectiveness of our proposed model.
>
>
>
> We hope that these additional responses help clarify your concerns. Please do not hesitate to let us know if there are any additional questions and we also appreciate your suggestions to further improve this submission.  Thank you.

---

> > ### Author Response · Authors · 2023-11-23
> > **A kind reminder**
> >
> > Dear reviewer bWo5,
> >
> > We hope that our additional responses clarify your concerns. We appreciate the opportunity to engage with you. As today is the last day of our discussion, please don't hesitate to share with us if you have any remaining questions or concerns; we will be happy to respond. Thank you.

---

> > > ### Comment · Reviewer_bWo5 · 2023-11-23
> > > **Re**
> > >
> > > Thank you for the feedback.
> > > I'm sorry that the current response did not address my concerns.
> > > Firstly, estimating the homophily ratio with less training set is indeed challenging. Additionally, in large-scale real-world datasets, many nodes lack labels (e.g., ogbn-papers100M), which further increases the difficulty of obtaining the homophily ratio. Secondly, more experimental supplements may be needed to complete the revised version of the paper. Taking into account the responses and other reviews, I decided to maintain the current score.

---

### Official Review · Reviewer_SYRx · 2023-10-31

**Soundness:** 3 good
**Presentation:** 3 good
**Contribution:** 3 good
**Rating:** 6
**Confidence:** 4

**Summary:**

Spectral Graph Neural Networks (GNNs) have become increasingly prevalent due to their strong performance in handling heterophily in graph data. However, optimal graph filters rely on a complex process and, to bypass this complexity, numerous polynomial filters have been proposed. These polynomial filters are designed to approximate the desired graph filters. A significant challenge arises because these polynomial methods mainly focus on specific types of graph structures. They struggle to accommodate graphs that display a diverse range of homophily and heterophily degrees.The paper aims to address this challenge by understanding the relationship between polynomial bases of designed graph filters and the diverse homophily and heterophily degrees in graphs. After the analysis, an adaptive heterophily basis is developed. The paper then integrates this with a homophily basis, leading to the creation of a universal polynomial filter known as "UniFilter".

Fundamentally, it seems that the adaptive basis ensures that the subsequent elements of the basis do not become too similar with higher k. The choice of this dissimilarity has been done in a very specific way, by computing signal specific basis vectors which are called heterophily basis in the paper. Unifilter is the combination of the standard polynomial basis with this heterophily basis. This combination is shown to give consistently good performance across varying range of homophily/heterophily datasets.

**Strengths:**

1. Interesting idea.
2. The results are very encouraging

**Weaknesses:**

1. The motivation for the choice of $\theta = \frac{\pi}{2}(1-h)$ from theorem 3, is not very straightforward and clear. The paper states that this choice is empirical, but there is very little given in terms of motivation for this exact form.
2. For this method, the knowledge of the homophily ratio seems to be important. In many practical scenarios, this may not be possible to be estimated accurately and even approximations could be difficult. No ablation study is presented showing the sensitivity of this model to the accurate knowledge of the homophily ratio.
3. The HetFilter seems to degrade rapidly past h=0.3 whereas OrtFilter is lot more graceful to the varying homophily ratio. It is unclear whether one would consider the presented fluctuations as inferior to the presented UniBasis. For UniBasis, in the region of h >= 0.3, the choice of tau should become extremely important (as is evident from Figure 4, where lower tau values can reduce performance on Cora by about 20 percentage points).

**Questions:**

Q1] Can you present a motivation for the choice of $\theta = \frac{\pi}{2}(1-h)$?
Q2] Imagine that we did not have the precise estimates of h, but we had approximate estimates of h with some variance. How much does the performance of the proposed approach change under this setting?
Q3] Since HetFilter can be expressed in terms of OrtFilter, There must be a w_k weights that should also work with OrtFilter. Then where is the gap in OrtFilter and HetFilter performance coming from?

---

> ### Author Response · Authors · 2023-11-18
> **Response to Reviewer SYRx (1/2)**
>
> Dear Reviewer SYRx,
>
> We sincerely thank you for your positive and valuable feedback on our paper. We address your comments and questions below. In the revised draft, we have updated the paper according to the reviewer's points. We mark our revision in $\textcolor{blue}{blue}$
>
>
>
> **W1**: The motivation for the choice of $\theta=\tfrac{\pi}{2}(1-h)$ from theorem 3, is not very straightforward and clear.
>
> **A1**: We would like to clarify that this setting is derived based on the insights from both Theorem 1 and Theorem 3.
>
> As we prove in Theorem 1, the frequency of desired signals is proportional to $1-h$. Therefore, the basis with its spectrum aligned with homophily ratios ($1-h$ in this case) enhances the capture of desired signals, leading to improved adaptability to the underlying graphs.
>
> As we prove in Theorem 3,  the expected signal frequency is monotonically increasing with $\theta$ for $\theta\in [0,\tfrac{\pi}{2})$. Combining with the conclusion in Theorem 1 and leveraging the monotonicity property, we thus empirically set $\theta=\tfrac{\pi}{2}(1-h)$. Consequently, setting $\theta=\tfrac{\pi}{2}(1-h)$ takes advantage of insights from both Theorem 1 and Theorem 3.
>
>
>
> As advised, we have updated the explanation in the revised version. Specifically, we revised Theorem 3 slight by explicitly pointing out frequency $f(\phi)=0$ for all-ones vector $\phi$ which is therefore served as the pivot vector. Then we have revised the explanation as follows, marked in $\textcolor{blue}{blue}$ in the updated version.
>
> "*Theorem 3 reveals the correlation between the expected frequency of the signal basis and its relative position to the $0$-frequency vector $\phi$ on regular graphs. This fact implicitly suggests that we may take the angles (relative position) between two basis vectors into consideration when aiming to achieve the desired basis spectrum on general graphs. Meanwhile, Theorem 2 discloses the growing similarity and asymptotic convergence phenomenon within the homophily basis. To mitigate this over-smoothing issue, we can intuitively enforce all pairs of basis vectors to form an appropriate angle of $\theta \in [0,\tfrac{\pi}{2}]$. Pertaining to this, Theorem 1 proves the spectral frequency of ideal signals proportional to $1-h$, aligning with the homophily ratios of the underlying graphs. By leveraging the monotonicity property proved in Theorem 2, we empirically set the $\theta:=\frac{\pi}{2}(1-h)$*."
>
>
>
>
>
> **W2**: For this method, the knowledge of the homophily ratio seems to be important. In many practical scenarios, this may not be possible to be estimated accurately and even approximations could be difficult. No ablation study is presented showing the sensitivity of this model to the accurate knowledge of the homophily ratio.
>
> **A2**: Please kindly refer to our global response to the homophily ratio issue. As demonstrated, this issue can be trivially resolved without compromising the performance of our model. As advised, we have added a discussion related to the homophily ratio and updated the experimental results in our revision.
>
> As suggested, we added an ablation study on the sensitivity of our proposed UniFilter to the estimated homophily ratio. Since UniFilter only utilizes the homophily basis on Cora ($\tau=1$ for Cora as shown in Table 5 in the Appendix in submission) and does not rely on the homophily ratio, we thus test UniFilter on Pubmed and Squirrel. To this end, we vary the estimated homophily ratio $\hat{h} \in \\{0.78, 0.79, 0.80, 0.81, 0.82\\}$ on Pubmed and  $\hat{h} \in\\{0.19,0.20,0.21,0.22,0.23\\}$ on Squirrel based on the results in Table C (in the global response). The achieved accuracy scores are presented in Table G.
>
> **Table G: Accuracy (%) for varying estimated homophily ratio $\hat{h}$ on Pubmed and Squirrel.**
>
> | Varying $\hat{h}$     | 0.78             | 0.79             | 0.80 (true $h$)     | 0.81             | 0.82             |
> | --------------------- | ---------------- | ---------------- | ------------------- | ---------------- | ---------------- |
> | Pubmed                | 91.03 $\pm$ 0.61 | 91.28 $\pm$ 0.64 | 91.34 $\pm$ 0.62    | 91.19 $\pm$ 0.67 | 91.17 $\pm$ 0.69 |
> | **Varying $\hat{h}$** | **0.19**         | **0.20**         | **0.21 (true $h$)** | **0.22**         | **0.23**         |
> | Squirrel              | 66.22 $\pm$ 1.43 | 66.96 $\pm$ 1.38 | 67.09 $\pm$ 1.08    | 67.01 $\pm$ 1.25 | 66.69 $\pm$ 1.26 |
>
> In Table G, UniFilter demonstrates consistent accuracy scores across different homophily ratios for both Pubmed and Squirrel. Notably, the accuracy variations remain minor, staying within a 1\% range from the scores under the true homophily ratio, particularly on the Pubmed dataset.
>
> Therefore, we observe that i) the homophily ratio can be effectively estimated using training sets, and ii) our proposed UniFilter is robust towards the variance of estimated homophily ratios.
>
> As advised, this ablation study is added in the Appendix in our revised version.

---

> > ### Author Response · Authors · 2023-11-18
> > **Response to Reviewer SYRx (2/2)**
> >
> > **W3**: The HetFilter seems to degrade rapidly past h=0.3 ... the choice of tau should become extremely important.
> >
> > **A3**: The results in Figure 3 aim to demonstrate the effectiveness and adaptability/universality of our proposed UniBasis. In particular, our model UniFilter employs UniBasis which combines the homophily basis and our designed heterophily basis with $\tau$ (defined in Equation (4)). To this end, we design three variants of Unifilter, named HetFilter, HomFilter, and OrtFilter. Specifically, HetFilter only utilizes the heterophily basis ($\tau=0$ for UniBais), while HomFilter only adopts the homophily basis ($\tau=1$ for UniBais). OrtFilter directly uses an orthogonal basis, a special heterophily basis by fixing $\theta=\tfrac{\pi}{2}$ without concerning the homophily ratio $h$.
> >
> > We evaluate the four models on the synthetic graphs across the varying homophily ratios. Figure 3 reports the accuracy gap which is the accuracy score of UniFilter minus the accuracy scores of the three variants respectively. The performance degradation of HetFilter for larger $h$ is due to that it only utilizes the heterophily basis, thus ineffective in handling homophily graphs as expected. Therefore, $\tau$ in UniBasis is supposed to align with the homophily ratio $h$ as well.
> >
> > The performance gap of OrtFilter indicates the advantage of our UniBasis over the orthogonal basis. Meanwhile, the corresponding fluctuations reveal the inferior adaptability of OrtFilter to graphs with varying homophily ratios.
> >
> >
> >
> >
> >
> > **Q1:** Can you present a motivation for the choice of $\theta=\tfrac{\pi}{2}(1-h)$?
> >
> > **A1**: Please kindly refer to our response to weakness 1.
> >
> >
> >
> > **Q2**: Imagine that we did not have the precise estimates of **h**, but we had approximate estimates of $h$ with some variance. How much does the performance of the proposed approach change under this setting?
> >
> > **A2**: Please kindly refer to our response to Weakness 2.
> >
> >
> >
> > **Q3**: Since HetFilter can be expressed in terms of OrtFilter, There must be a $w_k$ weights that should also work with OrtFilter. Then where is the gap in OrtFilter and HetFilter performance coming from?
> >
> > **A3**: HetFilter utilizes the heterophily basis by fixing $\theta=\tfrac{\pi}{2}(1-h)$ while OrtFilter employs an orthogonal basis with $\theta=\tfrac{\pi}{2}$.  According to linear algebra theory, a heterophily basis of a certain length theoretically provides the same expressive power as an orthogonal basis of identical length, as they both serve as bases for the same dimension of a vector space. This conclusion also applies to existing polynomial bases. However, in practical terms, different bases demonstrate varied empirical performance in node classification, exhibiting diverse outcomes for various graphs if offering no adaptability to the graphs.
> >
> > In our original submission, we discussed in **Section 4.3-Convergence Discussion** that the orthogonal basis achieves the maximum rate of approximation convergence. This intrinsic quality of orthogonal basis potentially equips OrtFilter with better adaptability compared to HetFilter. This may explain the performance gap in Figure 3. Nonetheless, as we explain in the response to Weakness 3, OrtFilter is inferior to our UniFilter in terms of effectiveness and adaptability.
> >
> >
> >
> > Thank you again for your valuable time and effort spent reviewing.

---

> > > ### Author Response · Authors · 2023-11-23
> > > **A kind reminder**
> > >
> > > Dear reviewer SYRx,
> > >
> > > We hope that you will find our responses satisfactory and that they help clarify your concerns. We appreciate the opportunity to engage with you. As today is the last day of our discussion, please don't hesitate to share with us if you have any remaining questions or concerns; we will be happy to respond. Thank you.

---

### Official Review · Reviewer_NCEm · 2023-11-01

**Soundness:** 1 poor
**Presentation:** 2 fair
**Contribution:** 2 fair
**Rating:** 3
**Confidence:** 4

**Summary:**

This paper introduces a new method called UniFilter for Spectral Graph Neural Networks (SGNNs) that addresses the issue of fixed polynomial filters in graph filters, accommodating the diverse heterophily degrees across different graphs. The core part of UniFilter is a vector basis called UniBasis, where the angle between each of two distinct basis vectors is $\theta=\frac{\pi}{2}(1-h)$.

The main flow that leads the authors to design UniBasis is as follows: First, the authors establish a theorem that depicts the correlation between homophily ratio $h$ and the frequency of a desired filtered vector signal. Next, the authors finds that on regular graphs, a signal's frequency is related to the its relative position towards the all-one vector.  This finding then leads the authors to build UniBasis.

In experiments, UniFilter show leading performances on real-world datasets compared with other state-of-the-art models.

**Strengths:**

- S1. The authors establish a theorem that depicts the correlation between homophily ratio $h$ and the frequency of the possibly desired output signal.
- S2. UniBasis is able to control the angle between each of the two basis vectors. Higher the homophily ratio,  smaller the angle.

**Weaknesses:**

- W1. The flow (as sketched in summary) lacks soundness.

  On regular graphs, the authors find that a signal's frequency is related to the its relative position towards the all-one vector. How does this observation leads to contraining the angles between basis vectors？ The authors roughly write: "... it explicitly prompts us to the potential correlation between the vector angles (relative position) and the basis spectrum."

- W2. $h$ is used as a prior knowledge to adjust the angles among basis vectors. However, the direct calculation of $h$ relies on labels on test sets. This issue is important since it is related to label leakage.

- W3. The claim "signals with **negative** weights are suppressed or eliminated as harmful information" lacks critical thinking. This assertion is related to the overall structure of the neural network, i.e., is there an neural layer after filtering?

**Questions:**

Please check weaknesses.

---

> ### Author Response · Authors · 2023-11-18
> **Response to Reviewer NCEm**
>
> Dear Reviewer NCEm,
>
> We sincerely thank you for your valuable feedback on our paper. We address your comments and questions below. In the revised draft, we have updated the paper according to the reviewer's points. We mark our revision in $\textcolor{blue}{blue}$.
>
>
>
> **W1**: The flow (as sketched in summary) lacks soundness.
>
> **A1**: We would like to clarify that this setting is derived based on the insights from Theorem 1, Theorem 2, and Theorem 3 combined together.
>
> We explain in detail as follows. First, Theorem 3 proves that the expected signal frequency is monotonically increased with the angle $\theta$ formed by the random signal $x$ and the all-ones vector $\phi$. Notice that the frequency of $\phi$ is $0$, i.e., $f(\phi)=0$, which thus is served as the pivot vector. This fact suggests us that we could take the angles (relative position) between two basis vectors into consideration when aiming to calibrate the basis spectrum on general graphs. Also, Theorem 2, the basis vectors in homophily basis exhibit growing similarity and would asymptotically converge (illustrated in Figure 1(a)), which means the angle formed by two consecutive homophily basis vectors becomes smaller until zero, resulting in the over-smoothing phenomenon. To mitigate this issue, it is a natural solution to fix the angle $\theta \in [0,\tfrac{\pi}{2})]$  between all pairs of basis vectors. Meanwhile, Theorem 1 proves the spectral frequency of ideal signals proportional to $1-h$. Inspired from the monotonicity relation disclosed in Theorem 3 and all the analyses above, we thus set $\theta=\tfrac{\pi}{2}(1-h)$ empirically.
>
> As advised, we have updated the explanation in the revised version. Specifically, we revised Theorem 3 slight by explicitly pointing out frequency $f(\phi)=0$ for all-ones vector $\phi$ which is therefore served as the pivot vector. Then we have revised the explanation as follows, marked in $\textcolor{blue}{blue}$ in the updated version.
>
> "*Theorem 3 reveals the correlation between the expected frequency of the signal basis and its relative position to the $0$-frequency vector $\phi$ on regular graphs. This fact implicitly suggests that we may take the angles (relative position) between two basis vectors into consideration when aiming to achieve the desired basis spectrum on general graphs. Meanwhile, Theorem 2 discloses the growing similarity and asymptotic convergence phenomenon within the homophily basis. To mitigate this over-smoothing issue, we can intuitively enforce all pairs of basis vectors to form an appropriate angle of $\theta \in [0,\tfrac{\pi}{2}]$. Pertaining to this, Theorem 1 proves the spectral frequency of ideal signals proportional to $1-h$, aligning with the homophily ratios of the underlying graphs. By leveraging the monotonicity property proved in Theorem 2, we empirically set the $\theta:=\frac{\pi}{2}(1-h)$*."
>
>
>
> **W2**: $h$ is used as a prior knowledge to adjust the angles among basis vectors. However, the direct calculation of $h$ relies on labels on test sets. This issue is important since it is related to label leakage.
>
> **A2**: Please kindly refer to our global response. As demonstrated, this issue can be trivially resolved without compromising the performance of our model. As advised, we have added a discussion related to the homophily ratio and updated the experimental results in our revision.
>
>
>
> **W3**: The claim "signals with **negative** weights are suppressed or eliminated as harmful information" lacks critical thinking. This assertion is related to the overall structure of the neural network, i.e., is there an neural layer after filtering?
>
> **A3**: We thank the reviewer for pointing it out. We would like to clarify that by following the convention in polynomial graph filters, the signals after filtering are fed into MLP in our model for weight learning. Therefore, the absolute values of learned weights signify the importance of the corresponding signal basis. Signals with negative weights in large absolute values are enhanced in the reverse direction. As advised, we have revised this sentence in our updated version as "signals with weights in large absolute values are enhanced while signals with small weights are suppressed".
>
>
>
> Thank you again for your valuable time and effort spent reviewing.

---

> > ### Comment · Reviewer_NCEm · 2023-11-20
> > **Response**
> >
> > **RA1**: How is **angles to the pivot vector** $\phi$ in regular graph related  to angles between basis vectors?
> >
> > **RA2**: Thanks for the implemented experiments. I will further discuss with the chair.
> >
> > **RA3**:
> > > ... the absolute values of learned weights signify the importance of the corresponding signal basis ...
> >
> > Large weights lead to small contributions when it is followed by a SoftMax layer, right?

---

> > > ### Author Response · Authors · 2023-11-20
> > > **Further responses**
> > >
> > > We thank the reviewer for the replies.
> > >
> > > **RA1**: How is **angles to the pivot vector** $\phi$ in regular graph related to angles between basis vectors?
> > >
> > > **Response**: Theorem 3 proves that the expected signal frequency of a random signal vector $x$ is monotonically increasing with the angle between $x$ and pivot vector $\phi$ on regular graphs. For pivot vector $\phi$, we have $f(\phi)=0$ for all graphs. Therefore, the frequency $f(x)$ of the signal vector $x$ is also the **relative frequency difference** between $x$ and $\phi$ since $f(\cdot)$ is non-negative for all $x\in \mathbb{R}^n$ (Lemma 2.1 in the submission).
> > >
> > > Theorem 3 might not be guaranteed for general graphs, but we get the inspiration from the above fact that the relative position (the angles) of two basis vectors, e.g. $x_1,x_2$ can be correlated to their relative frequency difference $|f(x_1)-f(x_2)|$ for general graphs. For example, if the two signal vectors $x_1,x_2$ are parallel, i.e., the angle equal to $0^\circ$, we have $f(x_1)=f(x_2)$; if the two signal vectors $x_1,x_2$ are orthogonal, i.e., the angle equal to $90^\circ$, the frequency difference $|f(x_1)-f(x_2)|$ can reach to the supremum value. Actually, only the orthogonal vector pair can reach the maximum frequency difference.
> > >
> > > Eventually, combining the conclusion from Theorem 2 to mitigate the over-smoothing and the conclusion from Theorem 1 to align with $1-h$, we thus fix the angles of all pairs of signal vectors to $\tfrac{\pi}{2}(1-h)$.
> > >
> > >
> > >
> > > **RA2**: Thanks for the implemented experiments. I will further discuss with the chair.
> > >
> > > **Response**: We sincerely appreciate your time and attention.
> > >
> > >
> > >
> > > **RA3**: Large weights lead to small contributions when it is followed by a SoftMax layer, right?
> > >
> > > **Response**: We thank the reviewer for the question. Before we continue, I would like to present the definition of weights here in case of any potential misunderstanding between us.
> > >
> > > The node representation $z$ is calculated as the weighted combination of signal basis, i.e., $\mathbf{z}=\sum^K_{k=0} \mathbf{w}_k P^kx$ (Equation (2) in the submission). Here $\mathbf{w} \in \mathbb{R}^{K+1}$ is the trainable weight vector and the $k$-th element $ \mathbf{w}_k$ is the weight value for the $k$-th signal vector. After that, $\mathbf{z}$ is fed into MLP, and then followed by a SoftMax layer for final prediction $y$, i.e.,   $y=\arg\max\\{\mathrm{Softmax}(\mathrm{MLP(\mathbf{z})})\\}$.
> > >
> > > Therefore, to our understanding, the larger weight $\mathbf{w}_k$ here enhances the corresponding signal vector $P^k x$ and thus contributes more to the node representation $\mathbf{z}$, thereby making more influence on the prediction results after a SoftMax layer.
> > >
> > >
> > >
> > > We hope that you will find these additional responses satisfactory and that they help clarify the concerns related to the angle design and the weight explanation. Please do not hesitate to let us know if there are any additional questions and we also appreciate your suggestions to further improve this submission. Thank you.

---

> ### Author Response · Authors · 2023-11-23
> **A kind reminder**
>
> Dear reviewer NCEm,
>
> We hope that our additional responses clarify your concerns. We appreciate the opportunity to engage with you. As today is the last day of our discussion, please don't hesitate to share with us if you have any remaining questions or concerns; we will be happy to respond. Thank you.

---

### Official Review · Reviewer_vpd8 · 2023-11-01

**Soundness:** 3 good
**Presentation:** 3 good
**Contribution:** 4 excellent
**Rating:** 6
**Confidence:** 2

**Summary:**

Learning on heterophilous graphs comes with underlying obstacles since most GNN models and spectral filters are based on homophily assumption. This paper expects to address this problem by designing a new graph filter combining both traditional homophilous bases and the proposed heterophilous bases. Specifically, the authors explore the correlation between homophily ratio and Euclidean space angles in spectral space, based on which the homophilous ratio-related bases can be established.

The experiments show the superiority of the proposed UniFilter on both homophilous datasets and heterophilous datasets. The analysis and ablation study strongly demonstrate the effectiveness of the proposed heterophilous bases which can adaptively capture useful heterophilous and homophilous information.

**Strengths:**

[Novelty] The main idea of designing homophilous-related bases is insightful and instructive. By investigating the correlation between homophily and bases, the well-designed heterophilous bases can adaptively and effectively address heterophilous graphs according to the homophilous ratio.
[Theoretical] The proposed UniFilter has strong theoretical support. It is partially guaranteed that the introduced heteraphilous filters can capture heterophilous information.
[Experiments] The  analysis of spectrum distribution of the learned frequencies clearly illustrated how the homophilous and heterophilous information is learned on different datasets.

**Weaknesses:**

1. [Related works] The paper loses investigations of the works also concentrating on heterophilous graphs [1-5]. The authors should compare these methods both experimentally as well conceptually, and explain the differences and relations. For example, [4] addresses heterophilous via combining different filters where each filter can be regarded as a basis, which is somehow similar to the proposed works.
2. [Completeness] This method will be effective under some assumptions, but the authors do not discuss the limitations. One example is as below.
3. [Theoretical] Theorem 3 shows the relationship between expectation and theta. However, the expectation is not accurate enough, especially when the distribution of spectra signal has a large variance, and at that time, constructing the basis according to theta would be invalid for capturing signals with extreme eigenvalue.

[1] Wang, T, et al. Powerful graph convolutional networks with adaptive propagation mechanism for homophily and heterophily. AAAI (2022)
[2] Ling. Y, et al. Dual Label-Guided Graph Refinement for Multi-View Graph Clustering. AAAI (2023).
[3] Chanpuriya, S.; and Musco, C. 2022. Simplified graph convolution with heterophily. NeurIPS 2022.
[4] Revisiting heterophily for graph neural networks. NIPS, 2022, 35: 1362-1375.
[5] Breaking the limit of graph neural networks by improving the assortativity of graphs with local mixing patterns. Proceedings of the 27th ACM SIGKDD Conference on Knowledge Discovery & Data Mining. 2021: 1541-1551.
[6] Is Homophily a Necessity for Graph Neural Networks?. ICLR. 2021.

**Questions:**

1. Please refer to weaknesses. especially weaknesses 3.

2. In Proof 3. authors claim that: "The negative value $\sum\frac{\lambda_i^{2k+1}(v_i^Tx_i)^2}{c1c2}$ decreases and the positive value $\sum\frac{\lambda_i^{2k+1}(v_i^Tx_i)^2}{c1c2}$" increases as the exponent k increases". How is this result derived? value range of $\lambda$ is $[-1,1]$, so the results should be the negative value decreases and the positive value decreases instead.

3. When connecting the bases with homophilous, the authors say "the basis spectrum is supposed to be aligned with homophily ratios" and "set $\theta:=\frac{\pi}{2}(1-h)$". I have two questions: 1) why does the basis spectrum need to align with homophily ratios? what is the advantage? and 2) why can it be aligned by setting $\theta:=\frac{\pi}{2}(1-h)$?

4. Could the proposed method mitigate the over-smoothing problem? Please include some experiments if possible.

---

> ### Author Response · Authors · 2023-11-18
> **Response to Reviewer vpd8 (1/3)**
>
> We sincerely thank you for your positive and valuable feedback on our paper. We address your comments and questions below. In the revised draft, we have updated the paper according to the reviewer's points. We mark our revision in $\textcolor{blue}{blue}$.
>
>
>
> **W1**: [Related works] The paper loses investigations of the works also concentrating on heterophilous graphs [1-5].
>
> **A1**: We appreciate the reviewer's suggestions. However, we would like to stress that the mentioned model in [4]  i.e., the ACM-GNN, is one of our baselines compared in Table 3 in our submission. Furthermore, we indeed conceptually discussed ACM-GNN [4] and the model WRGAT [5] in related work in our original submission.
>
> In this paper, we specifically focus on graph neural networks in the spectral perspective, therefore omitting the other three papers. By following your advice, we have added discussion on other missing papers in related works in our revised version. In Particular, HOG-GNN [1] designs a new propagation mechanism and considers the heterophily degrees between node pairs during neighbor aggregation, which is optimized from a spatial perspective. DualGR [2] focuses on the multi-view graph clustering problem and proposes dual label-guided graph refinement to handle heterophily graphs, which is a graph-level classification task. ASGC[3] simplifies the graph convolution operation by calculating a trainable Krylov matrix so as to adapt various heterophily graphs, which, however, is suboptimal as demonstrated in our experiments.
>
> Furthermore, we have included the model ASGC [3] and WRGAT [5] to compare with our method UniFilter by following their settings. The results are presented in Table D and Table E respectively as advised.
>
>
>
> **Table D: Accuracy (%) compared with ASGC [3] in the 60%/20%/20% data splits (settings in [3])**
>
> | Datasets  | Cora             | Citeseer         | Pubmed           | Actor            | Chameleon        | Squirrel         |
> | --------- | ---------------- | ---------------- | ---------------- | ---------------- | ---------------- | ---------------- |
> | ASGC [3]  | 85.35 $\pm$ 0.98 | 76.52 $\pm$ 0.36 | 84.17 $\pm$ 0.24 | 33.41 $\pm$ 0.80 | 71.38 $\pm$ 1.06 | 57.91 $\pm$ 0.89 |
> | UniFilter | 89.39 $\pm$ 1.36 | 81.61 $\pm$ 1.22 | 91.34 $\pm$ 0.62 | 41.32 $\pm$ 1.64 | 75.45 $\pm$ 0.74 | 67.09 $\pm$ 1.08 |
>
> **Table E: Accuracy (%) compared with WRGAT [5] in the 48%/32%/20% data splits (settings in [5])**
>
> | Datasets  | Cora             | Citeseer         | Pubmed           | Actor            | Chameleon        | Squirrel         |
> | --------- | ---------------- | ---------------- | ---------------- | ---------------- | ---------------- | ---------------- |
> | WRGAT [5] | 88.20 $\pm$ 2.26 | 76.81 $\pm$ 1.89 | 88.52 $\pm$ 0.92 | 36.53 $\pm$ 0.77 | 65.24 $\pm$ 0.87 | 48.85 $\pm$ 0.78 |
> | UniFilter | 88.94 $\pm$ 1.29 | 79.37 $\pm$ 1.36 | 90.19 $\pm$ 0.34 | 37.82 $\pm$ 1.04 | 74.46 $\pm$ 2.09 | 63.99 $\pm$ 1.41 |
>
> As shown, our UniFilter outperforms the two baseline models across all datasets, with notable advantages on some datasets. This observation further confirms the effectiveness of our proposed model.
>
> As advised, we have included those experimental results in Table 2 and Table 3 in our revision.
>
> [1] Wang, T, et al. Powerful graph convolutional networks with adaptive propagation mechanism for homophily and heterophily. AAAI (2022)
>
> [2] Ling. Y, et al. Dual Label-Guided Graph Refinement for Multi-View Graph Clustering. AAAI (2023).
>
> [3] Chanpuriya, S.; and Musco, C. 2022. Simplified graph convolution with heterophily. NeurIPS 2022.
>
> [4] Revisiting heterophily for graph neural networks. NIPS, 2022, 35: 1362-1375.
>
> [5] Breaking the limit of graph neural networks by improving the assortativity of graphs with local mixing patterns. Proceedings of the 27th ACM SIGKDD Conference on Knowledge Discovery & Data Mining. 2021: 1541-1551.
>
>
> **W2**: [Completeness] This method will be effective under some assumptions, but the authors do not discuss the limitations.
>
> **A2**: We thank the reviewer for the suggestion. Please kindly refer to our global response. As demonstrated in our global response, the assumption on heterophily ratios can be trivially avoided without compromising the performance of our proposed model. As advised, we have added a discussion related to the homophily ratio in our revision.

---

> ### Author Response · Authors · 2023-11-18
> **Response to Reviewer vpd8 (2/3)**
>
> **W3**: [Theoretical] Theorem 3 shows the relationship ... would be invalid for capturing signals with extreme eigenvalue.
>
> **A3**: First, we would like to clarify that (i) the expectation considered in Theorem 3 aims to explore a general relationship between signal frequency and its position by abstracting away the randomness of graphs, providing guidance to the design of our polynomial basis, and (ii) signals with extreme eigenvalue, i.e., high frequency or low frequency can be captured by our constructed basis. Specifically, the variance of the spectral frequency distribution for a random signal is the result of the randomness of graph structures. It affects how easily a signal is captured but does not affect if the graph signal is captured.
>
> Subsequently, we explain from both theoretical and practical perspectives. Theoretically, signals associated with eigenvalues are from a $n$-dimensional vector space where $n$ is the number of nodes in $G$. To capture any signals with any eigenvalues, our UniBasis can be constructed to contain $n$ basis vectors. Meanwhile, note that these basis vectors are *linear independent* since any two of them form the fixed $\theta$ angle in our design. In such a case, these $n$ linear-independent vector bases form a space basis, able to capture any signal vectors from that space with properly trained weights $w_k$, as well as the signals with extreme eigenvalue.
>
> Practically, as proved in Theorem 1, the frequency of desired signals is proportional to $1-h$ where $h$ is the homophily ratio. The signals with extreme eigenvalues (signals in either high frequency or low frequency) would be captured or enhanced if they approach the desired signal of graphs.  As shown in Figure 2 in our experiments, signals in both high frequency and low frequency are captured on corresponding datasets. Moreover, our UniFilter based on the UniBasis achieves SOTA performance on graphs with homophily ratios from $0.22$ to $0.81$.
>
>
>
> **Q1**: Please refer to weaknesses. especially weaknesses 3.
>
> **A1**: Please kindly refer to our response to weakness 3.
>
>
>
> **Q2**: In Proof 3. authors claim that: "The negative value $\sum \tfrac{\lambda^{2k+1}_i(v^T_i x_i)^2}{c1c2}$ decreases and the positive value $\sum \tfrac{\lambda^{2k+1}_i(v^T_id_i)^2}{c1c2}$" increases as the exponent k increases". How is this result derived? value range of $\lambda$ is [−1,1], so the results should be the negative value decreases and the positive value decreases instead.
>
> **A2**: We thank the reviewer for the careful checking. First, we would like to clarify that the value range of $\lambda$ is in ${\bf(-1,1]}$ ($\lambda=-1$ corresponds to bipartite graphs which are not considered in our paper) as stated in the proof, which is critical to deriving the result.
>
> Second, notice that for normalized vectors $\tfrac{P^k x}{\|P^k x\|}$ and $\tfrac{P^{k+1} x}{\|P^{k+1} x\|}$,  $\sum (\tfrac{\lambda^k_iv^T_ix}{c_1} )^2$ = $\sum (\tfrac{\lambda_i^{k+1} v^T_ix}{c_2} )^2$ =1 holds for all $k$ where $c_1=\|P^k x\|$ and $c_2=\|P^{k+1} x\|$. Since $v^T_i$ and $x$ remain unchanged,  values $(v^T_i x)^2$ are constant for all eigenvectors $v_i$, $i\in \\{1,2,\cdots,n \\}$. In the equation $\tfrac{P^k x \cdot P^{k+1} x}{\|P^k x\| \|P^{k+1} x\|}=\sum^n_{i=1} \tfrac{\lambda^{2k+1}_i(v^T_i x)^2}{c_1c_2}$, the exponent $2k+1$ of $\lambda_i$ keeps being odd integer. Therefore, $\tfrac{\lambda^{2k+1}_i(v^T_i x)^2}{c_1c_2}$ with $\lambda_i \in (-1,0)$ contributes the the negative part. $\tfrac{\lambda^{2k+1}_i(v^T_i x)^2}{c_1c_2}$ with $\lambda_i \in (0,1]$ corresponds the positive part.
>
> Notice that $c_1$ and $c_2$ are normalization factors, exerting equal scaling effects to both the negative part and the positive part. Their values also change accordingly along the increase of $k$ to ensure $\sum^n_{i=1} (\tfrac{\lambda^k_iv^\top_ix}{c_1} )^2 =\sum^n_{i=1} (\tfrac{\lambda^{k+1}_iv^\top_ix}{c_2} )^2=1$ for all $k$. As a consequence, we can infer that negative part $\tfrac{\lambda^{2k+1}_i(v^T_i x)^2}{c_1c_2}$  with $\lambda_i \in (-1,0)$ decreases while the positive part $\tfrac{\lambda^{2k+1}_i(v^T_i x)^2}{c_1c_2}$ with $\lambda_i \in (0,1]$ would increase along the increase of $k$.
>
> As advised, we have enhanced the proof in our revision.

---

> > ### Author Response · Authors · 2023-11-18
> > **Response to Reviewer vpd8 (3/3)**
> >
> > **Q3**: When connecting the bases with homophilous. I have two questions: 1) why does the basis spectrum need to align with homophily ratios? what is the advantage? and 2) why can it be aligned by setting $\theta=\tfrac{\pi}{2}(1-h)$?
> >
> > **A3**: We would like to clarify the two questions one by one as follows.
> >
> > **Response to question 1)**: As we prove in Theorem 1, the frequency of desired signals is proportional to $1-h$. Therefore, the basis with its spectrum aligned with homophily ratios ($1-h$ in this case) enhances the capture of desired signals (capture the signal more easily), leading to improved adaptability to the underlying graphs.
> >
> > **Response to question 2)**: As we prove in Theorem 3,  the expected signal frequency is monotonically increasing with $\theta$ for $\theta\in [0,\tfrac{\pi}{2})$. Combining with the conclusion in Theorem 1 and leveraging this monotonicity property, we thus empirically set $\theta=\tfrac{\pi}{2}(1-h)$. Consequently, setting $\theta=\tfrac{\pi}{2}(1-h)$ takes advantage of insights from both Theorem 1 and Theorem 3.
> >
> > As advised, we have updated the explanation in the revised version. Specifically, we revised Theorem 3 slight by explicitly pointing out frequency $f(\phi)=0$ for all-ones vector $\phi$, which is therefore served as the pivot vector. Then we have revised the explanation as follows, marked in $\textcolor{blue}{blue}$ in the updated version.
> >
> > "*Theorem 3 reveals the correlation between the expected frequency of the signal basis and its relative position to the $0$-frequency vector $\phi$ on regular graphs. This fact implicitly suggests that we may take the angles (relative position) between two basis vectors into consideration when aiming to achieve the desired basis spectrum on general graphs. Meanwhile, Theorem 2 discloses the growing similarity and asymptotic convergence phenomenon within the homophily basis. To mitigate this over-smoothing issue, we can intuitively enforce all pairs of basis vectors to form an appropriate angle of $\theta \in [0,\tfrac{\pi}{2}]$. Pertaining to this, Theorem 1 proves the spectral frequency of ideal signals proportional to $1-h$, aligning with the homophily ratios of the underlying graphs. By leveraging the monotonicity property proved in Theorem 2, we empirically set the $\theta:=\frac{\pi}{2}(1-h)$*."
> >
> >
> >
> > **Q4**: Could the proposed method mitigate the over-smoothing problem? Please include some experiments if possible.
> >
> > **A4**: Yes, our proposed method could address the over-smoothing problem. To elaborate, we prove in Theorem 2 that the homophily basis exhibits the *growing similarity* and *asymptotic convergence* properties, thus resulting in the over-smoothing issue (illustrated in Figure 1(a)). On the contrary, any two basis vectors in our proposed basis UniBasis form a fixed angle $\theta$. Therefore, they do not converge or become more similar along the increase of the number of basis vectors.
> >
> > For verification, we generate a $1000$-length of homophily basis and our proposed heterophily basis on dataset Squirrel respectively, i.e., consisting of $1000$ basis vectors. We calculate the degrees of the angle formed by all two consecutive basis vectors and present the degree distribution in Table F. Notice that the degree is averaged across the dimension of node features.
> >
> > **Table F: Degree ($^\circ$) distribution of the two consecutive basis vectors from the homophily basis and our basis on Squirrel ($h=0.22$)**
> >
> > | Basis             | $v_1, v_2$    | $v_2, v_3$    | $v_3, v_4$    | $v_4, v_5$    | $\cdots$ | $v_{996}, v_{997}$ | $v_{997}, v_{998}$ | $v_{998}, v_{999}$ | $v_{999}, v_{1000}$ |
> > | ----------------- | ------------- | :------------ | ------------- | ------------- | -------- | ------------------ | ------------------ | ------------------ | ------------------- |
> > | Homo. basis       | $88.74^\circ$ | $87.99^\circ$ | $87.76^\circ$ | $86.51^\circ$ | $\cdots$ | $0.0123^\circ$     | $0.0118^\circ$     | $0.0115^\circ$     | $0.0114^\circ$      |
> > | Our hetero. basis | $69.72^\circ$ | $70.03^\circ$ | $70.01^\circ$ | $70.05^\circ$ | $\cdots$ | $71.37^\circ$      | $71.23^\circ$      | $71.32^\circ$      | $71.22^\circ$       |
> >
> > As shown, degrees of the two consecutive basis vectors from the homophily basis approach to $0^\circ$, which indicates that the homophily basis on Squirrel converges asymptotically. On the contrary, degrees of our basis keep around $71^\circ$  determined by our setting $\theta=\tfrac{\pi}{2}(1-h)$ where $h=0.22$ for Squirrel (As we proved in the response to Weakness 2, $h$ can be effectively estimated using training data.). Notice that the variation of degrees in Table F is due to the average across $2089$-dimension features of Squirrel.
> >
> > As advised, we have included those experimental results in the Appendix in our revision.
> >
> > Thank you again for your valuable time and effort spent reviewing.

---

> ### Author Response · Authors · 2023-11-23
> **A kind reminder**
>
> Dear reviewer vpd8,
>
> We hope that you will find our responses satisfactory and that they help clarify your concerns. We appreciate the opportunity to engage with you. As today is the last day of our discussion, please don't hesitate to share with us if you have any remaining questions or concerns; we will be happy to respond. Thank you.

---

### Author Response · Authors · 2023-11-18
**Global Response (1/2)**

We sincerely appreciate all the reviewers for their insightful comments, which help us improve the quality and presentation of our work. All these updates have been highlighted in $\textcolor{blue}{\bf{blue}}$ in the revised revision.

We address one common question in this global response, while the remaining comments and specific questions are addressed in the individual response sections. To prevent any confusion between referencing tables in this rebuttal and those in the original submission, we have alphabetically numbered the tables in this rebuttal for clarity.

**Reviewer vpd8: W2 [Completeness] This method will be effective under some assumptions, but the authors do not discuss the limitations.**

**Reviewer NCEm: W2 The direct calculation of $h$ relies on labels on test sets.**

**Reviewer SYRx: W2 The knowledge of the homophily ratio seems to be important.**

**Reviewer bWo5: W1 The design of heterophilic bases relies on the dataset's homophily rate.**

**Response to Reviewers**: First, we acknowledge that our approach relies on the homophily ratio derived from the label sets of the entire graph, which are unknown in advance. However, we would like to stress that this challenge can be effectively addressed by estimating the homophily ratio from the labels of the training data, without compromising the efficacy of our model. The rationale is rooted in the sensible assumption that the distribution of the training data resembles the actual distribution of the testing data. Consequently, the estimated homophily ratio from the training data is a feasible proxy for the true homophily ratio. To confirm, we conducted experiments to validate the feasibility of the inferred homophily ratio and reevaluate our proposed spectral filter UniFilter.

For each dataset, we generate $10$ random splits of training/validation/testing are generated. For each split, we estimate the homophily ratio from the training data, using it as the input to UniFilter to construct polynomial bases for node classification. Notice that we estimate a new homophily ratio for each split and then obtain the corresponding accuracy score.

We present the estimated homophily ratio $\hat{h}$ with standard deviation averaged over the $10$ training sets for all datasets in Table A, along with the true homophily ratios for reference. Meanwhile, we also rerun UniFilter using the estimated homophily ratio from each split and report the averaged accuracy scores over the $10$ random splits on all datasets in Table B, along with the accuracy scores reported in the original submission for comparison. In particular, we denote the averaged homophily ratio estimations as $\hat{h}_1$ and $\hat{h}_2$ for data splits settings of 60%/20%/20% for polynomial filters (Table 2 in our submission) and 48%/32%/20% for model-optimized methods (Table 3 in our submission) respectively.

**Table A: Estimated homophily ratios in the two split settings and the true homophily ratios $h$.**

| Datasets                           | Cora            | Citeseer         | Pubmed           | Actor            | Chameleon       | Squirrel         |
| ---------------------------------- | --------------- | ---------------- | ---------------- | ---------------- | --------------- | ---------------- |
| True homo. ration $h$              | 0.81            | 0.73             | 0.80             | 0.22             | 0.23            | 0.22             |
| Estimated homo. ratio $\hat{h}_1$  | 0.82 $\pm$ 0.01 | 0.70 $\pm$ 0.01  | 0.79 $\pm$ 0.005 | 0.21 $\pm$ 0.004 | 0.24 $\pm$ 0.01 | 0.22 $\pm$ 0.005 |
| Estimated  homo. ratio $\hat{h}_2$ | 0.82 $\pm$ 0.01 | 0.69 $\pm$ 0.014 | 0.79 $\pm$ 0.01  | 0.21 $\pm$ 0.004 | 0.24 $\pm$ 0.01 | 0.22 $\pm$ 0.01  |

**Table B: Accuracy (%) of UniFilter using estimated homo. ratios under the two splits settings compared with the accuracy scores in original submission.**

| Datasets                                               | Cora             | Citeseer         | Pubmed           | Actor            | Chameleon        | Squirrel         |
| ------------------------------------------------------ | ---------------- | ---------------- | ---------------- | ---------------- | ---------------- | ---------------- |
| Accuracy in Table 2 in the original submission         | 89.39 $\pm$ 1.36 | 81.61 $\pm$ 1.22 | 91.34 $\pm$ 0.62 | 41.32 $\pm$ 1.64 | 75.45 $\pm$ 0.74 | 67.09 $\pm$ 1.08 |
| Accuracy rerun for Table 2 using estimated homo. ratio | 89.49 $\pm$ 1.35 | 81.39 $\pm$ 1.32 | 91.44 $\pm$ 0.50 | 40.84 $\pm$ 1.21 | 75.75 $\pm$ 1.65 | 67.40 $\pm$ 1.25 |
| Accuracy in Table 3 in the original submission         | 88.94 $\pm$ 1.29 | 79.37 $\pm$ 1.36 | 90.19 $\pm$ 0.34 | 37.82 $\pm$ 1.04 | 74.46 $\pm$ 2.09 | 63.99 $\pm$ 1.41 |
| Accuracy rerun for Table 3 using estimated homo. ratio | 89.12 $\pm$ 0.87 | 80.28 $\pm$ 1.31 | 90.19 $\pm$ 0.41 | 37.79 $\pm$ 1.11 | 73.66 $\pm$ 2.44 | 64.26 $\pm$ 1.46 |

---

> ### Author Response · Authors · 2023-11-18
> **Global Response (2/2)**
>
> To verify the estimation difficulty, we vary the percentages of the training set in {10%, 20%, 30%, 40%, 50%, 60%} on Cora and Squirrel, and then average the estimated homophily ratios $\hat{h}$ over 10 random splits. The results are presented in Table C.
>
> **Table C: Homophily ratio estimation $\hat{h}$ over varying training percentages on Cora and Squirrel**
>
> | Dataset  | 10%               | 20%               | 30%               | 40%               | 50%               | 60%              | True $h$ |
> | -------- | ----------------- | ----------------- | ----------------- | ----------------- | ----------------- | ---------------- | -------- |
> | Cora     | 0.83 $\pm$ 0.05   | 0.83 $\pm$0.04    | 0.83 $\pm$ 0.03   | 0.83 $\pm$ 0.01   | 0.82 $\pm$ 0.08   | 0.82$\pm$ 0.01   | 0.81     |
> | Squirrel | 0.23 $\pm$  0.014 | 0.22 $\pm$  0.011 | 0.22 $\pm$  0.010 | 0.22 $\pm$  0.006 | 0.22 $\pm$  0.005 | 0.22 $\pm$ 0.005 | 0.22     |
>
> As shown in Table A, the estimated values $\hat{h}_1$ and $\hat{h}_2$ derived from training datasets closely align with the actual homophily ratio $h$. There is a difference within a 2% range across all datasets, except for Citeseer. As shown in Table C, the estimated homophily ratio $\hat{h}$ is approaching the true homophily ratio $h$ across varying percentages of training data. This observation verifies that a high-quality estimation of the homophily ratio is accessible by the training set.
>
>
>
> We would like to emphasize that the accuracy scores based on estimated homophily ratios in Table B closely match those from the original submission, with a negligible difference of less than 1\%. As a consequence, our model UniFilter consistently outperforms all polynomial filters on the $6$ datasets except Actor and beats all model-optimized methods. This observation is consistent with the conclusion in the original submission. Therefore, the effectiveness of UniFilter is justified and the conclusions in the original paper still hold.
>
> As advised, we have clarified this homophily ratio issue and updated the experimental results based on the estimated homophily ratios in Table 2 and Table 3  in our revised version, marked in $\textcolor{blue}{blue}$.

---

### Author Response · Authors · 2023-11-21
**Revised version by incorporating Reviewers' Feedback Uploaded**

Dear Reviewers,

We thank the reviewers for the insightful feedback. Here we briefly summarize the changes to the updated version.

**Main text**

* (Section 4.2) We further enhanced the explanation of our design to constraint the angle $\theta$ between basis vectors as $\theta=\tfrac{\pi}{2}(1-h)$.
* (Section 4.2) We added a discussion on the homophily ratio estimation through training data instead of requiring the exact homophily ratio for our model. The performance of our model is not compromised by utilizing the estimated homophily ratios.

**Experiments**

* We updated our experimental results by utilizing the estimated homophily ratio. As shown, the performance of our model is consistent with the conclusion in the original submission. Therefore, the effectiveness of UniFilter is justified and the conclusions in the original paper still hold.
* We included two more baselines, i.e., ASGC [3] and WRGAT [5] in Table 2 and Table 3 respectively.

**Related work**

As advised, we have added and discussed the three papers [1,2,3] in related work. The suggested papers [4-5] are discussed in our original submission.

[1] Wang, T, et al. Powerful graph convolutional networks with adaptive propagation mechanism for homophily and heterophily. AAAI (2022)

[2] Ling. Y, et al. Dual Label-Guided Graph Refinement for Multi-View Graph Clustering. AAAI (2023).

[3] Chanpuriya, S.; and Musco, C. 2022. Simplified graph convolution with heterophily. NeurIPS 2022.

[4] Revisiting heterophily for graph neural networks. NIPS, 2022, 35: 1362-1375.

[5] Breaking the limit of graph neural networks by improving the assortativity of graphs with local mixing patterns. SIGKDD 2021.

**Appendix**

* (A.1 Proofs) We further clarified our proof for Theorem 2.
* (A.3 Additional Experiments) We included the experimental results of estimated homophily ratios across all tested datasets in Table 6.
* (A.3 Additional Experiments) We added an experiment to evaluate the difficulty of homophily ratio estimation by varying the percentages of training data. The results are presented in Table 7.
* (A.3 Additional Experiments) We added an experiment to assess the sensitivity of our model to the variance of the estimated homophily ratio. The results are presented in Table 8.
* (A.3 Additional Experiments) We added an experiment to demonstrate that our proposed model is able to mitigate the over-smoothing issue. The results are presented in Table 9.

**References**

The following three new references are added.

* [1] Wang, T, et al. Powerful graph convolutional networks with adaptive propagation mechanism for homophily and heterophily. AAAI (2022)

* [2] Ling. Y, et al. Dual Label-Guided Graph Refinement for Multi-View Graph Clustering. AAAI (2023).

* [3] Chanpuriya, S.; and Musco, C. 2022. Simplified graph convolution with heterophily. NeurIPS 2022.

---

### Meta-Review · Area_Chair_KEy2 · 2023-12-08

**Metareview:**

In this submission, the authors propose an adaptive spectral GNN, which adjusts polynomial filters according to the estimated homophily ratio. Experimental results show the potential of the proposed method in various learning tasks.

Strengths: The reviewers appreciate the idea proposed in this submission and think the theoretical part is interesting. They agree that this submission indeed has some merits worth to further investigating.

Weaknesses: (1) As a key step of the proposed method, homophily ratio estimation is often challenging in practical large-scale graphs with sparse labels. Although some existing methods have the potential to achieve such estimation and the authors made efforts to demonstrate the rationality of this step, this concern is not fully resolved in the rebuttal phase, and more analytic content should be added to the paper. (2) All reviewers have concerns about the writing and organization of this submission --- the connection between the theoretical results and the implementation is too loose, without sufficient and insightful explanations. For example, as the reviewers mentioned, the relation $\theta = \pi/2(1-h)$ is not justified strictly. Again, the authors' response did not resolve this concern completely.

In summary, although the submission is not qualified enough for ICLR at the current stage, I strongly encourage the authors to resubmit a new version to another venue.

**Justification For Why Not Higher Score:**

The idea of this work has some merits, but the implementation is questionable. Additionally, more analytic content is required for a new version.

**Justification For Why Not Lower Score:**

N/A

---

### Decision · Program_Chairs · 2024-01-16

Reject